# Quantifying the decay timescale of volcanic sulfur dioxide in the stratosphere

Paul A. Nicknish[1], Kane Stone[1], Susan Solomon[1], and Simon A. Carn[2]

[1]Department of Earth, Atmospheric, and Planetary Sciences, Massachusetts Institute of Technology, Cambridge, MA, USA
[2]Department of Geological and Mining Engineering and Sciences, Michigan Technological University, Houghton, MI, USA

**Correspondence:** Paul A. Nicknish (nicknish@mit.edu)

**Abstract.** The injection of sulfur dioxide ($SO_2$) into the stratosphere and its subsequent oxidation to form sulfate aerosols after large volcanic eruptions can have profound effects on Earth's climate. The removal of volcanic $SO_2$ in the stratosphere is thought to be driven by its gas-phase oxidation by the hydroxyl radical (OH); once oxidized, it goes on to form sulfate aerosols. However, it has also been suggested that heterogeneous oxidation on ash could also be important or even dominant, which would imply faster removal of $SO_2$ and thus faster formation of aerosols, at least in ash-rich plumes. Additionally, recent work uses an assumed exponential fit to determine the total $SO_2$ mass loading following large eruptions; the quality of this fit translates directly to the accuracy of the mass loading estimate. It is therefore of interest to examine how accurately the $SO_2$ decay timescale can be determined from observations and to compare observations to models. Here we evaluate the $SO_2$ decay timescale and its uncertainties following several significant eruptions using three different sets of satellite observations and compare these to the CESM2-WACCM6 model. We show that defining an accurate baseline against which a volcanic injection can be quantified increases the variability and uncertainty in the estimated decay timescale for some satellite data sets. While the typical decay timescale for $SO_2$ is on the order of a few weeks to a month, we find that uncertainties across different altitudes and eruptions results in timescales that can vary by more than a factor of 2. This makes it difficult to attribute variations in decay timescale to specific $SO_2$ removal processes for the events examined.

## 1 Introduction

The stratospheric aerosol layer is predominantly composed of sulfuric acid and water particles and plays a key role in both atmospheric chemistry and climate. These particles provide surfaces on which the activation of ozone-depleting chlorine takes place, and they can backscatter part of the incoming solar radiation out to space, moderating surface temperatures. Understanding the processes controlling stratospheric aerosol formation and residence times thus hinges on understanding the physical and chemical mechanisms influencing sulfur in the stratosphere. The most important sulfur-containing species for stratospheric aerosol formation is sulfur dioxide ($SO_2$). $SO_2$ is emitted naturally through volcanoes and anthropogenically via the combustion of fossil fuels and the smelting of sulfur-containing metal ore (Pumphrey et al., 2015).

An important source of stratospheric $SO_2$ in non-volcanic conditions is the photolysis of carbonyl sulfide (COS), which is the most abundant sulfur-containing gas in the atmosphere (Kremser et al., 2016). Important sources of COS include its direct flux

from the ocean, oxidation of marine-originating dimethyl sulfide and carbon disulfide, and direct and indirect anthropogenic emissions, among others (Kremser et al., 2016). With a tropospheric lifetime on the order of years (Brasseur and Solomon, 2005), COS can be transported to the stratosphere, where it then photolyzes and ultimately produces $SO_2$. Anthropogenic $SO_2$ can also frequently reach the stratosphere in the tropics via deep convection, especially in the Indian monsoon (Neely et al., 2014).

However, by far the most significant perturbations in stratospheric $SO_2$ are the result of moderate- to large-magnitude volcanic eruptions (volcanic explosivity index (VEI) 3+ and $\geq 1\,Tg\,SO_2$ emitted) (Solomon et al., 2011; Kremser et al., 2016; Schmidt et al., 2018; Carn et al., 2016). These Plinian-type eruptions feature plumes that entrain and heat ambient air as they rise, enhancing their own buoyancy and enabling them to ascend well above the tropopause (Carey and Bursik, 2000). Once the plume reaches the stratosphere, the $SO_2$ gas is oxidized and forms sulfate aerosols. The gas-phase oxidation process for
stratospheric $SO_2$ involves the following reaction sequence (Brasseur and Solomon, 2005):

$$SO_2 + OH + M \rightarrow HSO_3 + M \tag{R1}$$

$$HSO_3 + O_2 \rightarrow HO_2 + SO_3 \tag{R2}$$

$$SO_3 + H_2O \rightarrow H_2SO_4 \tag{R3}$$

where the initial oxidation of $SO_2$ by OH is the rate-limiting step. Depending on the environment in which the $H_2SO_4$ gas is present, it will either readily get taken up into pre-existing particles (increasing their size) or nucleate along with water vapor to form new particles (Yue, 1981). Once formed, the residence time of stratospheric sulfate aerosols ranges from a few months to a couple years and depends on the latitude, injection height, and time of year of the eruption. High latitude eruptions with
relatively low injection heights are associated with shorter residence times, whereas the aerosol cloud from tropical eruptions with high injection heights can persist for 1 to 2 years (Toohey et al., 2025; Myhre et al., 2013).

Given the far-reaching impacts of sulfur-derived stratospheric volcanic aerosols on both climate (e.g., McCormick et al., 1995; Robock, 2000; Stenchikov et al., 2009; Schmidt et al., 2018) and ozone ($O_3$) depletion (Solomon et al., 1998), quantifying the amount of $SO_2$ reaching the stratosphere following a major volcanic eruption and its subsequent chemical fate is
important (McKeen et al., 1984). In this work, our primary goal is to characterize the decay timescale $SO_2$ following large volcanic eruptions, which reflects the oxidation rate of $SO_2$. Moreover, since the rate-limiting step in the conversion of $SO_2$ to sulfate aerosol is the initial oxidation of $SO_2$, one would expect that the rate of sulfate formation should match the rate of $SO_2$ removal. Thus a better understanding of $SO_2$ removal times also translates to a better understanding of sulfate aerosol formation times, as discussed further below.

In addition to understanding the chemical fate of $SO_2$, an exponential fit of the $SO_2$ decay following large eruptions can be used to estimate the total stratospheric $SO_2$ mass burden. (Pumphrey et al., 2015; Höpfner et al., 2015). The $SO_2$ mass burden

is a key quantity for assessing the climate and chemical impacts of $SO_2$. Here we explain how its accuracy depends on the accuracy of the exponential fit.

Based on the gas-phase oxidation process given in the reactions in R1–R3, one would expect that the decay timescale of $SO_2$ increases with height due to the exponential decrease of pressure with height which limits the rate of reaction R1. Indeed, Carn et al. (2016), in their review of satellite measurements of volcanic degassing, show a substantial increase in total-column $SO_2$ $e$-folding time with injection height, ranging from less than a day for those eruptions that don't penetrate the tropopause to upwards of 40 days for the largest eruptions in the last hundred years (see their Fig. 14). However, total column measurements obscure vertical variations in $SO_2$ oxidation rates within the plume of a particular eruption; Höpfner et al. (2015) use vertically resolved observations from the Michelson Interferometer for Passive Atmospheric Sounding (MIPAS) and find that for a given eruption, the $SO_2$ decay timescale generally increases across 10 to 14 km, 14 to 18 km, and 18 to 22 km height bins (see their Table 3).

While these observations of the height dependence of the $SO_2$ decay timescale broadly match what we would expect based on the reactions in R1–R3, recent work has suggested other possible oxidation mechanisms for $SO_2$ in the stratosphere. Zhu et al. (2020) compared observed total column $SO_2$ decay timescales to model simulations following the 2014 eruption of Kelut in Indonesia. The eruption was notable in part because of a persistent layer of volcanic ash that remained for months after the eruption. They suggest that chemistry involving ash leads to a much shorter timescale of $SO_2$ (17 days when ash is included in their model versus 22–26 days with no ash; see their Table 1). However, Zhu et al. (2020) examined just one volcano, and the timescales they report fall within the range of values reported by Höpfner et al. (2015). Here we further examine available data on $SO_2$ decay following several different eruptions from different satellites, in part to test the potential role of ash on $SO_2$ oxidation.

## 1.1 Terminology

Before preceding with the rest of the paper, we think it is useful to clarify precisely what we are calculating with respect to the removal of $SO_2$ from the stratosphere. As discussed thoroughly in recent work by Toohey et al. (2025), terms such as "residence time," "$e$-folding time," and "decay timescale" are often taken to be synonymous, when in fact they refer to related but distinct quantities.

Following the discussion in Toohey et al. (2025), consider a tracer injected into a reservoir at time $t = 0$. The mean residence time is the average of the residence times for the tracer particles leaving the reservoir. The $e$-folding time is the time required for the concentration of the tracer in the reservoir to reach $1/e$ of its initial value. When the removal of the tracer is a pure exponential decay, the residence time will equal the $e$-folding time. However, both of these quantities can be defined, regardless of the functional form of the decay; for non-exponential decay, they are not necessarily equal.

In this paper we focus on the "decay timescale" of $SO_2$, which can be defined as follows. Let $W(t)$ be the mass of the tracer in the reservoir after time $t = 0$. The decay timescale $\tau_d$ is defined as

$$\tau_d = -\left[\frac{\mathrm{d}}{\mathrm{d}t}\ln\left(W(t)\right)\right]^{-1}. \tag{1}$$

In other words, it is the reciprocal of slope of $\ln(W(t))$ when plotted against time. For a perfectly exponential decay of the tracer mass in the reservoir, the plot of $\ln(W(t))$ vs time will be a straight line, and the slope (the decay timescale) will be equal to both the $e$-folding time and the mean residence time. However, for non-exponential decay, the decay timescale will vary in time. In practice, we calculate the decay timescale via the slope of a best fit line to the natural log of $SO_2$ mass. The median decay timescale during the decay period is taken as a measure of the overall behavior and for comparison with other results

in the literature. Furthermore, we include a bounds indicating the spread of decay timescales throughout the decay period. In general, the tighter the bounds, the closer the decay is to exponential. Wider bounds are indicative of increased variability and uncertainty, and suggests that a single, calculated $e$-folding time does not sufficiently capture the true nature of the decay. The lack of consensus on this terminology in the literature adds a layer of complication when comparing results. One of our goals here is to draw attention to this potential source of uncertainty.

## 2   Data and methods

### 2.1   Satellite observations

Satellite observations of volcanic $SO_2$ emissions are provided by both nadir-viewing and limb-sounding instruments (e.g., Carn et al., 2016). Nadir measurements using backscattered ultraviolet (UV) radiation (e.g., OMI and OMPS) measure the total vertical column density (VCD) of $SO_2$. They provide good horizontal resolution ($\sim$10–50 km) and contiguous, global coverage

but no vertical resolution (vertical profile information can sometimes be derived from UV measurements, but it is computationally expensive and not a standard product). Nadir observations by infrared (IR) sounders have similar horizontal resolution and can be used to retrieve $SO_2$ altitude (e.g., Clarisse et al., 2014), but this requires relatively high $SO_2$ column amounts and hence is problematic for dispersed volcanic plumes. In contrast, limb-sounding instruments using emitted microwave and IR radiation (e.g., MLS and MIPAS ) provide vertically resolved $SO_2$ profiles (with vertical spacing of $\sim$1.5–3 km) but they have

low horizontal resolution ($\sim$170 km for MLS and 420 km for MIPAS) and only measure the stratospheric $SO_2$ contribution to the column.

The retrieval of $SO_2$ VCD from nadir UV measurements requires an assumed $SO_2$ vertical profile as input, introducing uncertainty if the assumed profile differs from the actual vertical distribution. However, limb-sounding instruments measure the $SO_2$ vertical profile directly, thus eliminating some of the uncertainty in $SO_2$ mass loading inherent to nadir measurements.

Furthermore, because limb-sounding instruments detect emitted radiation from a long horizontal (or tangent) path through the atmosphere, they provide greater sensitivity to volcanic $SO_2$ when the gas is highly dispersed in thin, horizontally extensive layers. Compared to nadir UV measurements, microwave and IR limb-sounders also provide higher sensitivity to $SO_2$ at very high latitudes.

Nadir observations are therefore optimal for measuring $SO_2$ mass loading in recently erupted volcanic $SO_2$ plumes (within

hours to a few days after eruption), which are relatively compact and poorly sampled by limb sounders. However, once the $SO_2$ becomes dispersed in the atmosphere, the greater sensitivity of limb sounders is advantageous for monitoring the decay of stratospheric $SO_2$, especially at high latitudes. Eruptions at mid- to high-latitudes typically experience stronger wind shear

than tropical eruptions (e.g., due to interaction with the polar jet stream) and hence may have a tendency to disperse more quickly below the detection limit of nadir instruments.

Here, we focus on data from two limb-sounding instruments (MIPAS and MLS), and one nadir instrument (OMI). These instruments are discussed in more detail in the following sections.

### 2.1.1    MIPAS

The Michelson Interferometer for Passive Atmospheric Sounding (MIPAS) instrument was an atmospheric limb sounder that measured radiation in the region 685–2410 $cm^{-1}$ via a Fourier transform spectrometer (Fischer et al., 2008). The instrument
was on the polar orbiting satellite Envisat and operated from 1 March 2002 until 8 April 2012. During the time period of the retrievals used in this work, the spectral resolution was 0.0625 $cm^{-1}$, the horizontal resolution was 420 km, and the vertical resolution was approximately 1.5 km (Höpfner et al., 2015). We accessed the data at https://www.imk-asf.kit.edu/english/308.php and used the reduced spectral resolution version of the data (which is valid from 2005 to 2012). In addition to $SO_2$ volume mixing ratio, we use retrieved pressure and temperature to convert the volume mixing ratio to mass, as described in later sections.
As suggested within the MIPAS data files, we select valid data by only using points where visibility = 1 and akm_diagonal > 0.03.

### 2.1.2    MLS

The Microwave Limb Sounder (MLS) is an instrument on NASA's Aura satellite, which launched in July 2004 and has a sun-synchronous orbit (Waters et al., 2006; Schoeberl et al., 2006). The instrument measures thermal emission in the microwave
from Earth's limb and has done so with little interruption from August 2004 to the time of writing (Pumphrey et al., 2015). The retrievals of temperature and $SO_2$ mixing ratios used in this work are reported on pressure levels with an approximate spacing of 3 km, and the horizontal resolution is approximately 170 km (Livesey et al., 2022). We use Level 2 Version 5 (V5) daily swath $SO_2$ mixing ratio data, accessed at https://disc.gsfc.nasa.gov/datasets/ML2SO2005 (Read and Livesey, 2021). The V5 data features minor changes from previous versions, including improved cloud detection, changes in the calculation of $O_3$
and carbon monoxide (stated to have secondary impacts on $SO_2$), and updates to the handling of background radiance signals (Livesey et al., 2022). This data is obtained via the 240 GHz radiometer on the MLS instrument (Pumphrey et al., 2015). In addition to the $SO_2$ mixing ratio, this data set reports the temperature at each pressure level, and we use this in our calculation of $SO_2$ mass and altitude above sea level.

The MLS documentation highlights that the retrieval algorithm can generate negative mixing ratios, and the correct way
to deal with these is to average over a sufficiently large horizontal area (Livesey et al., 2022). We apply all of the suggested masking for the data given in Livesey et al. (2022), and we average our data over 10° latitude bands. Even after masking and averaging, negative mixing ratios are prominent in the MLS data, particularly lower in the atmosphere.

### 2.1.3 OMI

Like MLS, the Ozone Monitoring Instrument (OMI) is also on NASA's Aura satellite. With only minor gaps since August 2004, OMI measures ultraviolet and visible nadir solar backscatter (Levelt et al., 2006) and thus does not provide high-resolution vertical resolution for $SO_2$. Here we use the Level 2, Version 3 stratospheric estimate of $SO_2$ VCD (Li et al., 2020). These estimates of VCD are reported in the variable ColumnAmountSO2_STL from the OMI product and are given in Dobson units ($1\,DU = 2.69 \times 10^{16}\,molec\,cm^{-2}$). We follow the guidelines for flagging erroneous values given in the documentation for the data set (see Li et al. (2020)). OMI's stratospheric estimate of VCD is derived using an assumed lower-stratospheric $SO_2$ profile with a center of mass at 18 km. Note that results using the OMI output in ColumnAmountSO2_TRU, which assumes a center of mass at 13 km, had little impact on our results. See Li et al. (2017) for more details on the retrieval algorithm.

### 2.2 Model Data

We compare the satellite data to model output from the Whole Atmosphere Community Climate Model version 6 (WACCM6), which is a component of the Community Earth System Model version 2 (CESM2). Data for 1980–2014, which we use here, were initially published in Gettelman et al. (2019). See this paper for a thorough overview on the details of the model. Throughout the paper, we refer to the model output simply as WACCM.

The horizontal resolution of the model is 1.9° latitude by 2.5° longitude. There are 88 levels in the vertical, extending up to approximately 140 km above the surface of the Earth. Vertical resolution in the upper troposphere and stratosphere is 1–2 km. All major volcanic eruptions from 1980 to 2014 are included in the model. The chemical mechanism includes a detailed representation of stratospheric chemistry, and importantly, the oxidation of stratospheric $SO_2$ is driven by the gas-phase reaction with OH only. Thus, the model provides a useful baseline for determining other potential oxidation pathways in the observations.

### 2.3 Calculation of $SO_2$ mass

The method used to calculate the $SO_2$ stratospheric burden depends on the units of the initial $SO_2$ data. When starting with $SO_2$ volume mixing ratio (as is the case for MIPAS, MLS, and the WACCM data), we first convert the volume mixing ratio to a mass mixing ratio using the molar masses of $SO_2$ and air. For every vertical profile, pressure and temperature from the respective product are then used to calculate the air density at each pressure level (using the ideal gas law). Multiplying the air density with the mass mixing ratio yields at the density of $SO_2$ at each pressure level.

The MIPAS data is reported on an altitude coordinate with a spacing of 1 km (between 10 km and 22 km), and the MLS and WACCM data are reported on a pressure coordinate. In order to compare the data sets, we interpolate the MLS and WACCM data to the altitude coordinate from MIPAS.

For the MLS data, this is done in several steps. At every vertical profile, we first calculate the change in altitude $\Delta z_i$ between successive points $p_i$ and $p_{i+1}$ on the MLS pressure coordinate using hydrostatic balance:

$$\Delta z_i = z_{i+1} - z_i = \int\limits_{p_i}^{p_{i+1}} -\frac{1}{g\rho_{\text{air}}}\mathrm{d}p \qquad (2)$$

In practice, we evaluate this integral discretely and assume that the density $\rho_{\text{air}}$ varies linearly from $\rho_{\text{air}}(p_i)$ to $\rho_{\text{air}}(p_{i+1})$. The maximum value in the MLS pressure coordinate is $1000\,\text{hPa}$. By assuming that this pressure corresponds to an altitude of $0\,\text{km}$, we can use equation (2) to calculate the altitude of every pressure level of every vertical profile in MLS. With this, we then linearly interpolate each vertical profile from MLS to the altitude coordinate of the MIPAS data.

For the WACCM data, the model output geopotential height is used for the interpolation to the MIPAS height grid.

Once the data is on an altitude vertical coordinate, we integrate in the vertical and horizontal to get the total mass of $SO_2$. As we are interested in the oxidation of $SO_2$ as a function of height, we group our data into three height bins used by Höpfner et al. (2015) (10–14 km, 14–18 km, and 18-22 km) and vertically integrate in each height bin. We also calculate the total $SO_2$ mass in the upper troposphere/lower stratosphere by integrating from 10 to 22 km. Given the variation in tropopause height with latitude, the 10 to 14 km and 14 to 18 km layers won't necessarily be entirely in the stratosphere in low latitudes (Hoffmann and Spang, 2022). However, we use the vertical divisions here for consistency with past work, and future work could consider a vertical division based on tropopause height.

Following Höpfner et al. (2015), for the horizontal integration, we group our data into $10°$ latitude bands, take the mean of the vertically integrated $SO_2$ data falling in each band, and multiply by the area of the respective latitude band. We perform this calculation for each of the three height bins to find the mass of $SO_2$ in each height bin within each latitude band. Finally, for each height bin, we sum up the mass of $SO_2$ in all of the latitude bands that were clearly affected by the volcanic eruption.

The $SO_2$ mass from OMI is calculated by horizontally integrating the reported vertical column density values. The horizontal integration is the same as that described for the MLS and MIPAS data.

## 2.4 Calculation of the decay timescale

We determine the decay timescale by first calculating the perturbation due to the eruption from a background $SO_2$ level that occurs in the absence of any volcanic influence. We determine the background for each height bin and $10°$ latitude band. The time period that defines the the non-volcanic background varies by eruption and product, but in general, we select a background such that the $SO_2$ perturbation decays to 0 given enough time. For some eruptions and products, care needs to be taken to select an appropriate background; for example, there is a signal from the eruption of Okmok in the MIPAS data prior to the Kasatochi eruption, and thus we select a background period long after the influence of either eruption is seen in the time series (see Fig. 3f). For the MLS data, we also remove the apparent seasonal cycle for each height bin (if the cycle is present; see Sect. 4.2 for more details on how we remove the seasonal cycle, and the implications of doing so).

Once the volcanic $SO_2$ perturbation is isolated, we then determine the decay timescale by fitting a best fit line of the form (using the notation of Höpfner et al. (2015))

$$M_{\Delta h_i}(t) = M_{\Delta h_i}(t_0) \exp\left(-\frac{t - t_0}{\tau_{\Delta h_i}}\right), \qquad (3)$$

where $M_{\Delta h_i}$ is the mass of $SO_2$ in height bin $\Delta h_i$, $\tau_{\Delta h_i}$ is the decay timescale in that bin, and $t_0$ is the start of the window used for the calculation. Volcanic plumes can have strong impacts on the local chemical environment in the stratosphere by, for example, altering OH concentrations (especially for high $SO_2$ concentrations), photolysis rates, and water vapor content (McKeen et al., 1984; Carn et al., 2022; Glaze et al., 1997). These impacts can in turn influence the oxidation rate of $SO_2$, and in particular lead to variations in the decay timescale during the decay of the $SO_2$ burden. Thus, in order to assess this potential variability, we perform the calculation over a series of moving windows in time. We use windows of three lengths: 15, 20, and 25 days. For each window length, the calculation is repeated for five different windows with the first day in each window spaced five days apart. The start of the first window coincides with when the $SO_2$ mass begins to decline. When the $SO_2$ perturbation decays in a period shorter than five windows, we adjust the number of windows appropriately. Figure 1 illustrates the way in which the windows are varied to calculate the decay timescale.

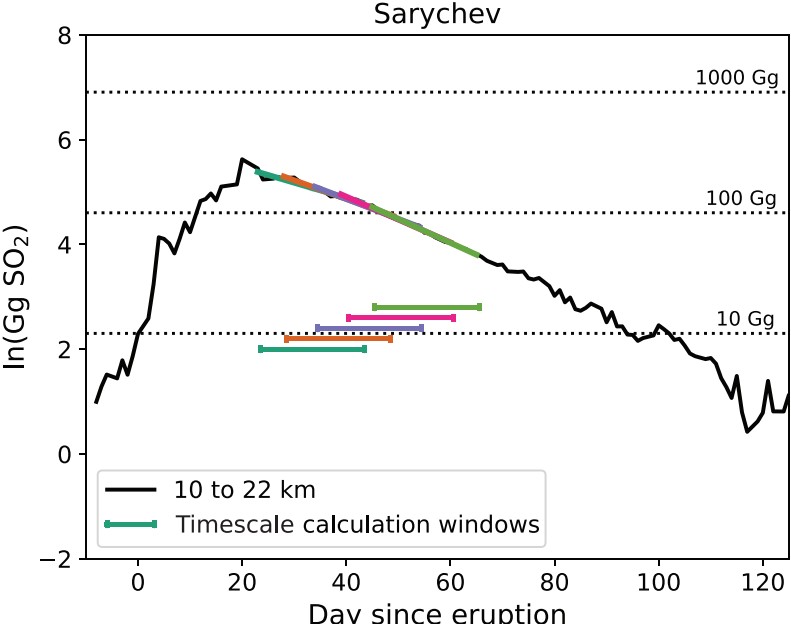

**Figure 1.** Visual example of the windows used to calculate the decay timescale throughout the decay period. This plot shows the $SO_2$ mass after the 2009 Sarychev eruption between 10 and 22 km from MIPAS (solid black line). The colored lines are linear fits of the curve during different periods of the decay, as indicated by the horizontal line of the same color below the black curve. The window length in this example is 20 days.

The decay timescales collected from all the windows (all lengths and times) give a sense of the total variability of the oxidation rate of the volcanic $SO_2$. The median and 5th and 95th percentiles are used to quantify the spread. Note we use the median, as opposed to the mean, to limit the influence of the tail towards long timescales.

## 3   Timescales of $SO_2$ removal and sulfate aerosol formation

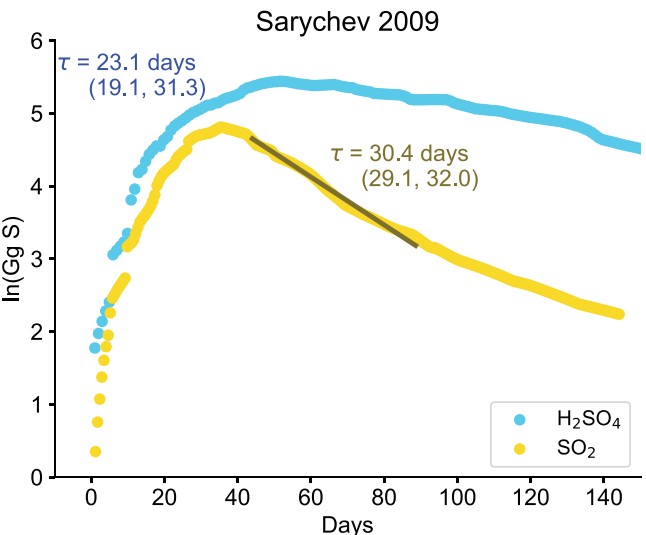

**Figure 2.** Comparison of $SO_2$ decay timescales and sulfate formation timescales. Data is from Günther et al. (2018) and is retrieved from MIPAS. Masses shown are the total mass from 10.5 to 22.5 km. The timescale of sulfate formation is comparable to that of $SO_2$ removal. Note that time series here are in units of Gg sulfur (as opposed to Gg $SO_2$ used in other plots in the paper). Values in parentheses denote 5th and 95th percentiles.

The oxidation and subsequent removal of stratospheric $SO_2$ is directly related to the formation of climate- and chemistry-

altering sulfate aerosols. Thus, a central, underlying motivation for this work is that better quantification of the $SO_2$ removal process directly translates to a deeper understanding of aerosol formation. Before proceeding with the rest of the analysis, we present simple comparison between the timescale of $SO_2$ removal (which is determined by the oxidation rate) and the timescale of sulfate aerosol formation for the 2009 Sarychev eruption (Fig. 2). Assuming that

1. the sulfate aerosol perturbation is entirely the result of chemical conversion from $SO_2$

2. the timescale of sulfate aerosol removal is significantly greater than that of sulfate aerosol formation

3. the loss of $SO_2$ (and subsequent conversion to sulfate aerosol) is a first-order reaction with timescale $\tau$

then the growth of the sulfur mass contained in the sulfate aerosol ($m_{S,aero}$) is given by

$$\frac{dm_{S,aero}}{dt} = -\frac{dm_{S,SO_2}}{dt} = \frac{M_0}{\tau} \exp\left(\frac{-t}{\tau}\right) \tag{4}$$

where $M_0$ is a constant and $m_{S,SO_2}$ is the mass of sulfur contained in $SO_2$. Solving for $m_{S,aero}$ as a function of time gives

$$m_{S,aero}(t) = M_0 \left(1 - \exp\left(\frac{-t}{\tau}\right)\right). \tag{5}$$

Here $M_0$ is taken to be the observed peak in the sulfur mass contained in the sulfate aerosol.

Using sulfate data from Günther et al. (2018), we estimate the timescale $\tau$ using Eq. (5). We subtract the observed time series of sulfur mass contained in sulfate from its observed maximum. We then take the natural log of the resulting curve and use a linear fit to estimate $\tau$. This fit is done between the start of the eruption and the peak in the sulfate sulfur mass. By varying the window used for this fit, we obtain 5[th] and 95[th] percentiles for the timescale as a confidence interval around the median.

We compare this timescale derived from sulfate formation to that derived from a linear fit to the natural log of the mass of sulfur contained in $SO_2$. The two median timescales are reasonably similar (23.1 vs 30.4 days), and there is some overlap between the 5[th]-to-95[th] percentile ranges, which gives confidence to the notion that a better understanding of volcanic $SO_2$ decay timescales will improve our understanding of sulfate aerosol formation.

The decay time estimate obtained from Eq. (5) is an approximation and will be sensitive to the instrument's ability to measure sulfate aerosol (see Günther et al. (2018) for further discussion on the sulfate retrieval). Furthermore, once the sulfate aerosol is formed, it will be removed via sedimentation and transport by the large-scale circulation (e.g., the Brewer-Dobson circulation). Generally these processes are much slower than the timescale for sulfate production (which motivates assumption 2 above). Nonetheless, they may influence the shape of sulfate perturbation (the blue scatter points in Fig. 2) and the subsequent timescale estimate. For a more thorough discussion of the different timescales involved in sulfate aerosol formation and residence time, see Toohey et al. (2025).

## 4 Determining the $SO_2$ decay rate in limb-sounding observations

### 4.1 Comparing MIPAS and MLS

We use vertically resolved retrievals of $SO_2$ from two limb-sounding products, the Microwave Limb Sounder (MLS) and the Michelson Interferometer for Passive Atmospheric Sounding (MIPAS) (see Sect. 2 for more details). Both instruments have high temporal and near global coverage, making their observations suitable for assessing the decay of $SO_2$ both vertically and horizontally as the volcanic plume is mixed and dispersed following the eruption (Höpfner et al., 2015). While past work has analyzed $SO_2$ retrievals in each product individually (for MLS see Pumphrey et al. (2015); for MIPAS see Höpfner et al. (2015) and Günther et al. (2018)), to our knowledge this is the first direct comparison of their vertically resolved $SO_2$ measurements. The Aura satellite, which carries the MLS instrument, is expected to last until 2028, while MIPAS operated from 2002 through 2012. We focus our analysis on large (greater than $1\,Tg\,SO_2$ emitted, Carn et al. (2016)) eruptions covered by both instruments.

Figure 3 shows the observations from MLS and MIPAS from 2008. While there were a few smaller eruptions in that year, by far the most significant was the eruption of Kasatochi on August 7, 2008 (set to day 0 in the figure and indicated with a vertical dashed line). An island volcano in the Aleutian Islands (52.12°N, 175.51°W), Kasatochi injected an estimated $2\,\mathrm{Tg}\,SO_2$ into the atmosphere, with the plume reaching a height of $15\,\mathrm{km}$ (Carn, 2024). The top row of Fig. 3 shows the $SO_2$ daily mean volume mixing ratio (in ppb) in the lower stratosphere as a function of time and latitude retrieved from MLS (left) and MIPAS (right). MLS data is reported on a pressure vertical coordinate, whereas MIPAS has altitude as the vertical coordinate. The data plotted in Fig. 3a and 3b shows the daily-mean volume mixing ratio at a similar distance above the surface of the Earth prior to any sort of interpolation or integration. This highlights the inherent differences in the underlying data between the two products. While the eruption is clearly visible in both data sets, as indicated by the elevated $SO_2$ shortly after the eruption (day 0), the mixing ratios reported by MLS are negative for much of the year, whereas in MIPAS they are positive and much closer to 0 (compare Fig. 3a to Fig. 3b). Negative mixing ratios are unphysical and an artifact of the MLS retrieval algorithm; nonetheless, the MLS documentation recommends including them in subsequent calculations (Livesey et al., 2022). However, aggregating these negative mixing ratios over a large geographic area (e.g., 40°N to 90°N) and converting to mass results in large negative values that complicate the interpretation of the data, but are nonetheless included here.

Figures 3c and 3d show the total $SO_2$ mass between 40°N and 90°N for three different height bins, 10 to $14\,\mathrm{km}$, 14 to $18\,\mathrm{km}$, and 18 to $22\,\mathrm{km}$ (see Sect. 2.3 for more details). We choose these bins to be consistent with those used in Höpfner et al. (2015). While the Kasatochi eruption is clearly visible in Fig. 3c, the MLS data in the 10 to $14\,\mathrm{km}$ bin shows large negative masses throughout the year. Additionally, the MLS mass in the 10 to $14\,\mathrm{km}$ and 14 to $18\,\mathrm{km}$ bins feature a seasonal cycle with an amplitude much larger than what is expected for background stratospheric $SO_2$ (Pumphrey et al., 2015; Höpfner et al., 2013). The background $SO_2$ values in this region of the stratosphere are on the order of a few tens to a hundred ppt (see Fig. 7 in Höpfner et al., 2013), whereas the seasonal cycle shown here and in Pumphrey et al. (2015) have an amplitude on the order of ppb. Furthermore, the amplitude of the MLS seasonal cycle is too large to be explained by other potential sources of stratospheric sulfur such as the annual flux of OCS into the stratosphere (Karu et al., 2023). Pumphrey et al. (2015) noted this unrealistic seasonal cycle in MLS $SO_2$ when using Version 2 of the data and attributed it to interference from $O_3$ and nitric acid ($HNO_3$), as these species exhibit strong emission lines in the passband of the radiometer used to measure $SO_2$. The implications of this seasonal cycle are discussed further in Sect. 4.2. We also note that at low altitudes, the noise of the MLS data likely arises at least in part from the fact that microwave emission lines—and thus the signal received by the instrument—are subject to pressure broadening. This may explain the reduction of noise seen in the 18 to $22\,\mathrm{km}$ height bin in MLS, as the impacts of pressure broadening decrease with height (Pierrehumbert, 2010).

# Kasatochi

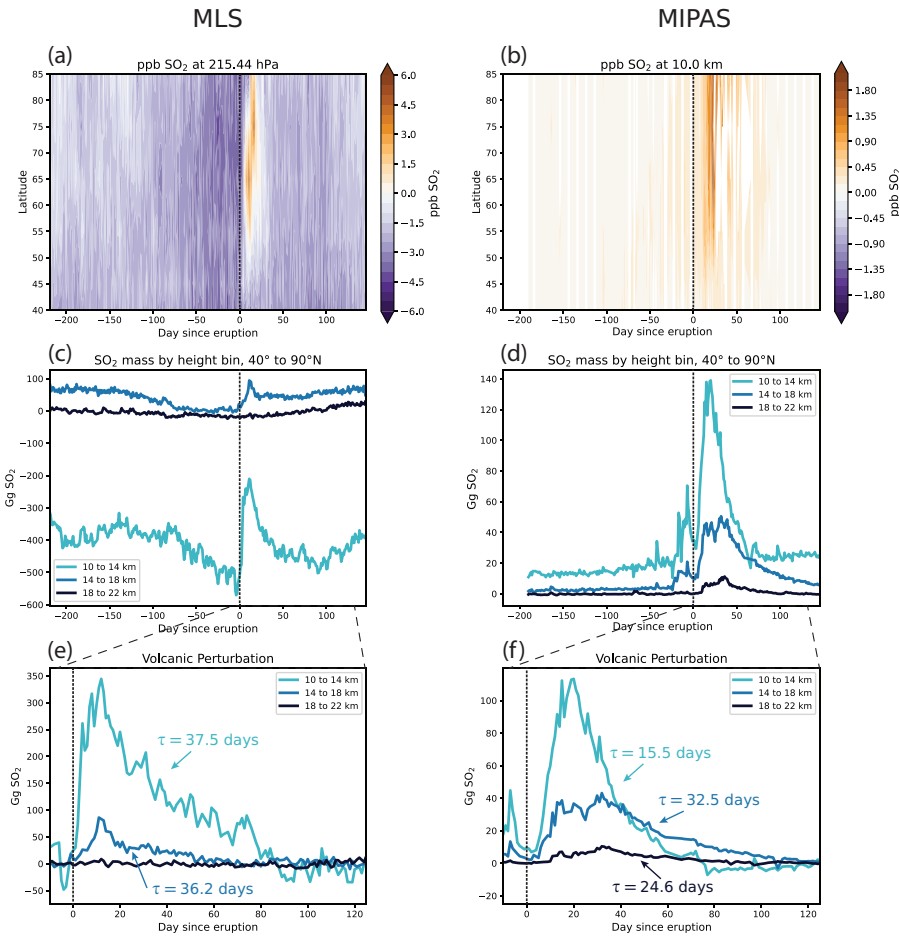

**Figure 3.** Comparison of MLS and MIPAS observations during 2008 and the Kasatochi eruption. MLS observations are shown in the left column, and MIPAS observations are shown in the right column. In all panels day 0 corresponds to the start of the Kasatochi eruption (August 7, 2008; Carn (2024)), as denoted by the vertical dashed line. The top row shows the volume mixing ratio in ppb as a function of latitude and time for the (a) MLS and (b) MIPAS at comparable altitudes on the native height coordinate of the respective observational product. Note the different colorbar scale between the two panels. The mixing ratio values have been averaged on each day in 10° latitude bands but are otherwise unmodified from those reported in the respective data sets. The middle row shows the mass of $SO_2$ in Gg over the course of the year in the 10 to 14 km, 14 to 18 km, and 18 to 22 km height bins. The bottom row shows the perturbation of $SO_2$ in each height bin resulting from the Kasatochi eruption and the associated decay timescales. There was no significant volcanic signal for the MLS data in the 18 to 22 height bin, and thus no values are reported. Note that in panels (c) and (e), the MLS data has been interpolated to the height coordinate of MIPAS.

Compared to MLS, the MIPAS data is easier to interpret. There is no significant seasonal cycle in the background, and the large injection of $SO_2$ due to Kasatochi is clearly seen (Fig. 3d). Note that the spike in $SO_2$ at lower altitudes prior to the Kasatochi eruption is likely due to the more minor eruption of Okmok in Alaska on July 12, 2008 (53.42°N, 168.13°W, $0.15\,Tg\,SO_2$ emitted; Carn (2024)). The signal from this eruption is not clear in the MLS data.

However, the peak perturbation of $SO_2$ after the Kasatochi eruption is roughly a factor of 3 greater in the MLS data than in the MIPAS data; this is because MIPAS does not fully sample the volcanic plume at its most dense. As noted by Höpfner et al. (2015), the difference between MIPAS and MLS in the initial part of the plume's dispersion and decay is thought to be the result of both interference from volcanic particles and saturation in the spectral lines measured by the MIPAS instrument in the presence of very high $SO_2$ concentrations (such as those seen immediately after a large eruption). Thus, there are advantages and disadvantages to both satellite sensors in this application. MLS provides a better measurement of the peak input of $SO_2$ relative to MIPAS data. However, despite these shortcomings, the decay of the $SO_2$ mass is clearly seen in MIPAS once the initial plume has dispersed enough for the signal to reach the instrument, i.e., when the $SO_2$ radiance is no longer saturated. Furthermore, it is worth noting that the magnitude of the MLS values in Version 5 of the data are roughly a factor of 4 smaller than those in Version 2 reported by Pumphrey et al. (2015). The reason for this remains unclear, and no explanation or documentation for this difference was found in the literature.

We performed the same evaluation for the eruptions of Sarychev in 2009 (48.01° N, 153.20° W, $1.2\,Tg\,SO_2$ emitted) and Nabro in 2011 (13.37° N, 41.70° E, $1.975\,Tg\,SO_2$ emitted), the two other major eruptions covered by both MLS and MIPAS (Carn, 2024). The results for Sarychev and Nabro are shown in Fig. A1 and Fig. A2, respectively. In both cases we see negative mixing ratios and masses in the lower two height bins in the MLS data. For Sarychev, the magnitude of the perturbation is comparable in MLS and MIPAS, with MIPAS showing around $100\,Gg$ more mass in the 10-14 km height bin (Fig. A1e and Fig. A1f). Previously, Höpfner et al. (2015) indicated that MLS displayed more mass than MIPAS after Sarychev (their Fig. 13), and the change that we see here may be due to differences between Version 2 and Version 5 of the MLS data. Furthermore, in the case of the Nabro eruption, the MLS data in the 10 to 14 km height bin does not show any signal of the eruption. In the MIPAS data for Nabro, there is an upward trend in $SO_2$ in the 10 to 14 km bin prior to the eruption. This could possibly be linked to upward transport of $SO_2$ in the Asian summer monsoon (Neely et al., 2014), as Nabro is a tropical eruption, and thus the latitude bands influenced by the eruption are also more likely to be influenced by the monsoon.

## 4.2 Background seasonal cycle in the MLS data

As shown in Fig. 3c, there is an apparent seasonal cycle in the mass of $SO_2$ measured by MLS in both the 10 to 14 km and 14 to 18 km height bins that is not realistic (Pumphrey et al., 2015; Höpfner et al., 2013). While Pumphrey et al. (2015) speculate that this is due to leakage of information from $O_3$ and $HNO_3$, it is not obvious what the shape of the seasonal cycle should be, and we incorporate the resulting decay timescales corresponding to different possible seasonal cycles into an overall estimate of uncertainty in the decay timescale.

In order to sample possible shapes of the seasonal cycle, at each height bin we first take a 35-day running mean of the MLS time series with the non-volcanic background removed (see Sect. 2.4 for details on how we define the non-volcanic

background). We then replace the values from day $M$ to day $N$ (where we vary $M$ from -5 to 1 and $N$ from 50 to 105 in the
case of Kasatochi; we adjust the range of values for $M$ and $N$ for each eruption so that the seasonal cycles are reasonable)
with a line between the $SO_2$ mass on day $M$ and the $SO_2$ mass on day $N$. In other words, $M$ and $N$ define the section that we
remove from the time series prior to determining the seasonal cycle, and varying these parameters results in different cycles.
We then use a discrete fast Fourier transform to filter out the high frequency variability, leaving us with the seasonal cycle.
Once we determine this, we subtract out the seasonal cycle from the time series.

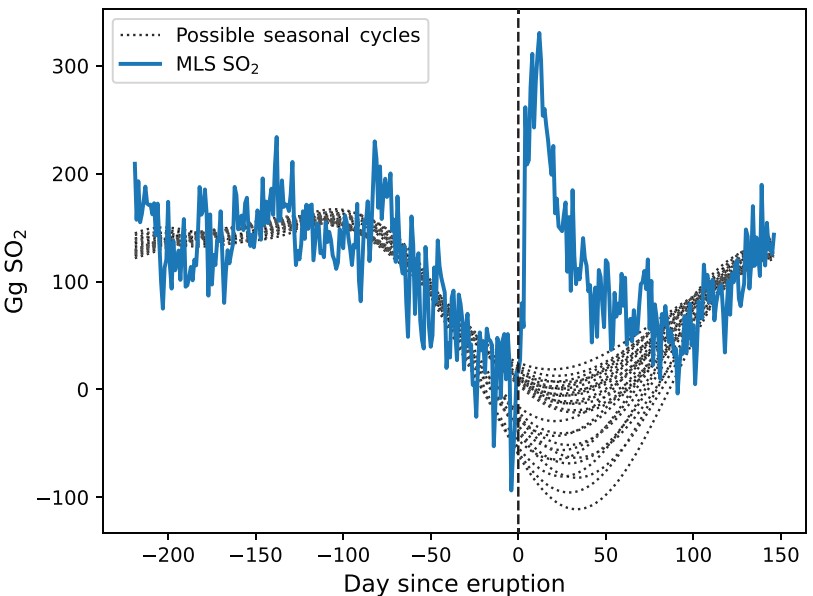

**Figure 4.** Possible background seasonal cycles in the MLS data. The blue curve shows MLS mass in the 10 to 14 km height bin following
Kasatochi. Note that a constant has been subtracted from the time series shown in Fig. 3c so that the curve is roughly zero immediately
before the eruption. The dotted black lines show possible seasonal cycles that could have potentially occurred in the absence of the Kasatochi
eruption. As in Fig. 3, day 0 corresponds to the eruption of Kasatochi and is indicated by the dashed vertical line.

Figure 4 demonstrates the possible seasonal cycles in the MLS data derived using the method outlined above. The figure
uses data from the Kasatochi eruption in the 10 to 14 km height bin, but the resulting spectrum of seasonal cycles is similar
across eruptions and height bins. The presence of the seasonal cycle leads to a range of possibilities in the shape and decay of
the volcanic perturbation, and this ultimately translates into greater uncertainty when deriving a decay timescale from the MLS
data (see Sect. 4.3 and Table 1 for more details).

The approach outlined above samples possible seasonal cycles using the observed time series in a given year of an eruption.
One could also potentially infer the interfering seasonal cycle from years without large volcanic eruptions; however, this

presents its own challenges. There is a 100 to 200 Gg spread in $SO_2$ mass on any given day of the year in non-volcanic conditions (Fig. S1). Using the mean across years for the background seasonal cycle could result in an offset of $\pm50$ to 100 Gg

from the actual disturbing seasonal cycle. This difference is large enough to impart a significant bias in the estimated decay rate, thus limiting our ability to constrain the uncertainty. The seasonal cycle and inter-seasonal variability in MLS $SO_2$ is interesting in its own right: it warrants further investigation and could potentially be calibrated based on the observed amounts of $O_3$ and $HNO_3$, but this is beyond the scope of the current paper.

### 4.3    Comparing $SO_2$ decay timescales for different volcanoes

We compare decay timescales and their estimated uncertainties for MLS and MIPAS in Table 1 for the three largest eruptions during the time period covered by both products: Kasatochi in 2008, Sarychev in 2009, and Nabro in 2011. These eruptions all occurred in local summer. However, Kasatochi and Sarychev are both high latitude, northern hemisphere eruptions, whereas Nabro is a tropical eruption. Differences in tropopause height, OH concentrations, and local dynamics between the tropics and higher latitudes make comparisons between Nabro and the high latitude eruptions difficult. In particular, the tropopause in the

tropics during the local summer is around 16 km, whereas that for the high latitudes in the northern hemisphere is closer to 11 km (Hoffmann and Spang, 2022). As such, the majority of the three layers considered in this analysis are likely to be in the stratosphere for the Kasatochi and Sarychev eruptions. After the Nabro eruption, likely only the 18 to 22 km layer was initially fully in the stratosphere; however, the plume was quickly advected to higher latitudes—where the tropopause is lower—by the Asian Monsoon anticyclone in just a few days (Clarisse et al., 2014). There are large vertical gradients in $H_2O$ and therefore

OH at the tropopause (Milz et al., 2005; Jiang et al., 2015), and thus whether a layer falls above or below the tropopause will potentially significantly impact $SO_2$ removal times. A more precise treatment of calculating $SO_2$ decay based on whether a measurement falls above or below the tropopause is left for future work.

Despite this variability between different eruptions, the comparison between the observational products for a given eruption is illustrative of the uncertainty associated with each product. These values are summarized in Table 1. Values in bold indicate

the median decay timescale from a collection of values calculated by varying the time window used for the linear fit of the exponential decay. Our uncertainty values (shown in parentheses) are the 5th and 95th percentiles, which gives a sense of the variability in perturbation decay rate depending on how it is calculated. See Sect. 2.4 for more details on this calculation. Given the range of methods used to calculate the decay rate of volcanic $SO_2$ in the literature (e.g., Höpfner et al., 2015, Table 3 and references therein), our intent with these uncertainty bounds is to demonstrate how a spread in the decay timescale can arise

by changing the time period used for the calculation. Plume-induced changes in chemistry is one potential mechanism driving this behavior (McKeen et al., 1984). Vertical transport by the background circulation of the stratosphere is unlikely to have a significant impact on our results as it is quite slow—on the order of tenths of $\mathrm{mm\,s^{-1}}$ or hundreths of $\mathrm{km\,day^{-1}}$—compared to the timescale of $SO_2$ decay (Butchart, 2014). Khaykin et al. (2022) did report an unusual radiative self-lofting of the Raikoke volcanic plume in 2019; the observed vertical ascent for this eruption was upwards of $2\,\mathrm{mm\,s^{-1}}$ ($0.17\,\mathrm{km\,day^{-1}}$) and would

be fast enough to impact our results. This phenomenon has not been noted for any of the volcanoes examined here, though it is a potential source of uncertainty and worth examining in future work.

In general there is greater uncertainty in the MLS-derived decay timescales, in part because the ambiguity of the seasonal cycle present in the data is compounded with the variability that arises in changing the decay time window. The volcanic signals in the MIPAS data are clearer, and uncertainty in the decay timescale here stems from uncertainty in the exponential fit of the

$SO_2$ mass following the eruption alone. These results suggest that MIPAS, as an infrared sounder, can provide better estimates of the decay timescale throughout the depth of the atmosphere compared to MLS. However, we emphasize again that MIPAS does not capture the peak input as well as MLS.

**Table 1.** Comparison of calculated decay timescales between this work and Höpfner et al. (2015). For the MIPAS, MLS, and WACCM rows, bold values are the median decay timescale, and the values in parentheses are the 5[th] and 95[th] percentiles. See Sect. 2.4 and Sect. 4.2 for more details on how these are calculated. MIPAS-derived values from Höpfner et al. (2015) are indicated by MIPAS H2015. Dashes indicate a lack of a clear signal for that product at that height bin. All timescales have units of days.

### Kasatochi
August 7, 2008; (52.12°N, 175.51°W)

|                      | 10 to 14 km          | 14 to 18 km          | 18 to 22 km          |
| -------------------- | -------------------- | -------------------- | -------------------- |
| **MIPAS (this study)** | **15.5** (11.9, 17.9) | **32.5** (30.0, 37.9) | **24.6** (19.7, 33.0) |
| **MIPAS (H2015)**    | **14** (13, 15)      | **23** (18, 28)      | **32** (28, 36)      |
| **MLS**              | **37.5** (22.4, 67.1) | **36.3** (12.8, 55.8) | —                    |
| **WACCM**            | **17.5** (14.9, 21.7) | **23.8** (15.1, 29.3) | **22.5** (19.6, 23.4) |

### Sarychev
June 15, 2009; (48.01° N, 153.20° W)

|                      | 10 to 14 km          | 14 to 18 km          | 18 to 22 km          |
| -------------------- | -------------------- | -------------------- | -------------------- |
| **MIPAS (this study)** | **25.4** (22.5, 36.6) | **30.7** (23.6, 60.5) | **38.4** (26.4, 65.9) |
| **MIPAS (H2015)**    | **15** (13, 17)      | **25** (24, 26)      | **38** (36, 40)      |
| **MLS**              | **10.7** (6.6, 27.2) | **22.0** (10.6, 32.9) | —                    |
| **WACCM**            | **10.4** (10.1, 12.1) | **14.1** (13.6, 20.8) | —                    |

### Nabro
June 13, 2011 (13.37° N, 41.70° E)

|                      | 10 to 14 km          | 14 to 18 km          | 18 to 22 km          |
| -------------------- | -------------------- | -------------------- | -------------------- |
| **MIPAS (this study)** | **16.6** (10.4, 30.1) | **29.5** (24.8, 37.2) | **35.6** (32.5, 56.4) |
| **MIPAS (H2015)**    | **11** (8, 14)       | **23** (21, 25)      | **27** (26, 28)      |
| **MLS**              | —                    | **26.5** (15.4, 66.8) | —                    |
| **WACCM**            | **9.2** (6.3, 17.4)  | **10.4** (10.0, 10.6) | **19.3** (17.8, 26.2) |

Table 1 also compares our calculated decay timescales with those calculated by Höpfner et al. (2015) using MIPAS retrievals. Note that Höpfner et al. (2015) refers to these values as "lifetimes" and calculates them by varying the window for the fit of a straight line to the natural logarithm of the $SO_2$ mass. They use longer windows (around 30 days) than we use here. Their values show a clear increase of decay timescale with height, which is generally also seen in our results (with the exception of some of the values for Kasatochi). Additionally, the $5^{th}$ to $95^{th}$ percentile ranges reported here are large, and in nearly all cases the results reported in Höpfner et al. (2015) fall within the ranges given here.

We also compare the observations from the satellite products with output from the WACCM model (Gettelman et al., 2019) for these three eruptions (see Table 1 and Fig. 5). The WACCM model only incorporates gas-phase oxidation of $SO_2$ and thus provides a useful baseline to use in comparing the real world measurements.

The expected increase in decay timescale with height based on gas-phase oxidation alone is most obvious in the Nabro eruption for the WACCM data. In the satellite data, this is most clearly seen for MIPAS in the Sarychev and Nabro eruptions. The median WACCM decay timescales (in which OH is the only oxidative agent) tend to fall below the median decay timescales from the satellite observations. The one notable exception is the Sarychev eruption in the 10 to 14 km height bin, where the median decay timescales between MLS and WACCM are quite close at around 11 days. However, it is challenging to make further conclusions from this alone due to the previously discussed issues in the MLS data. For Nabro, the median decay timescale from WACCM is outside the uncertainty range of the observations for all three height bins. There are a variety of potential causes for the discrepancy between the model and observations. One could be differences in water vapor and between the model and observations. However, comparisons between WACCM and MLS water vaper generally show strong agreement, and it is not clear that this should be the main culprit (Froidevaux et al., 2019). Model treatment of the interplay between particle scattering and photolysis inside the volcanic plume initially after the eruption could also play a role, but we do not explore this discrepancy further here.

While the observed median decay timescale is generally on the order of 10 to 40 days, there is a wide range in values across the different products and height bins. The uncertainty in obtaining the decay timescale in the WACCM model is quite low, as it only arises due to uncertainty in the fitting of the decay. This decay is—as one would expect for model output—quite clear and easy to quantify; moreover, the tight bounds suggest that the removal of $SO_2$ in the model is indeed exponential. However, uncertainties in the process understanding of OH chemistry in the model are likely to be a larger source of error; this likely also limits the interpretation of any discrepancies in data/model comparisons. While MIPAS displays smaller uncertainties than MLS (Table 1), on the whole the observations have a much wider range of decay timescales. For example, the observational decay timescales in the 10 to 14 km height bin for Sarychev span from roughly 36 to under 7 days. For Kasatochi, the observational decay timescales in the 10 to 14 km height bin range from over 67 to under 12 days. It is also worth noting that previous estimates of decay timescales (derived using a variety of satellite products) for these eruptions do fall within the (rather large) uncertainty ranges shown here (see Höpfner et al. (2015) Table 3 and references therein).

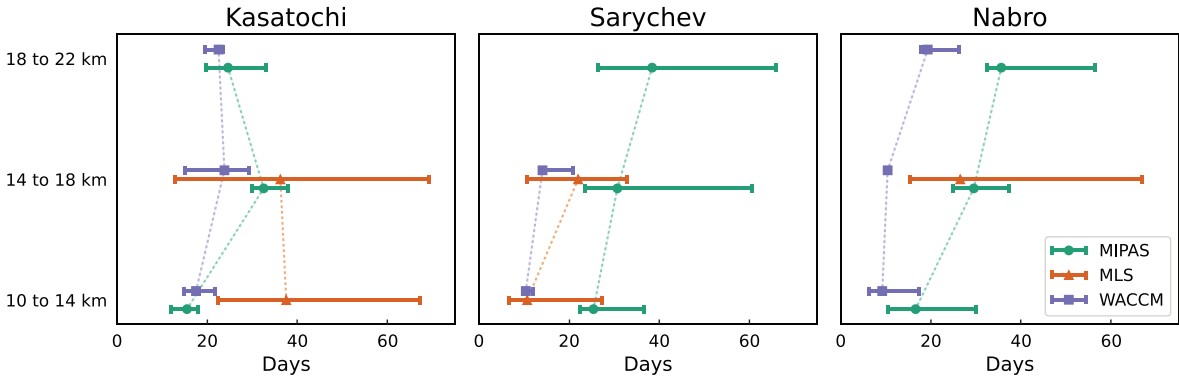

**Figure 5.** SO$_2$ decay timescale in the 10 to 14 km, 14 to 18 km, and 18 to 22 km height bins for Kasatochi 2008, Sarychev 2009, and Nabro 2011. Markers indicate the median decay timescale, and error bars show the 5[th] and 95[th] percentile. For a description of how the median and error bars are calculated, see Sect. 2.4 and Sect. 4.2 . A missing data point for a given product and height bin indicates the lack of a sufficient signal for the calculation.

## 5  Decay timescales for total SO$_2$ between 10 and 22 km and comparisons with OMI

We compare the results from MLS, MIPAS, and WACCM with SO$_2$ retrievals from the Ozone Monitoring Instrument (OMI). OMI is a popular choice in recent work examining the decay of SO$_2$ following eruptions (e.g., Carn et al., 2022; Zhu et al., 2020; Krotkov et al., 2010), and we include an analysis of it here for a comparison of how limb-sounders and nadir-sounders capture the removal of volcanic SO$_2$. Note that OMI is just one of several nadir-sounding instruments that measure SO$_2$, and a detailed comparison with other instruments (such as AIRS) is left to future work.

Unlike MLS and MIPAS, OMI is a nadir-viewing instrument that measures backscattered ultraviolet and visible radiation (Li et al., 2017). OMI reports the vertical column density of SO$_2$ and lacks the detailed vertical resolution for SO$_2$ provided by MLS and MIPAS. However, OMI is able to provide estimates of SO$_2$ within different vertical layers of the atmosphere by assuming a vertical SO$_2$ profile and iteratively adjusting it to fit the observed backscattered radiation. Here we use the stratospheric data set from OMI, which is intended for studying explosive volcanic eruptions. For more details see Sect. 2.1.3 and Li et al. (2017).

Since OMI only provides a total column perspective of the stratosphere, we compare the decay of the volcanic perturbation in OMI with that between 10 and 22 km in the MLS, MIPAS, and WACCM products. Figure 6 shows this comparison for the Kasatochi eruption. The OMI-derived decay timescale of 7.3 days is substantially faster than that in either of the other two satellite products or the WACCM model. Indeed, the 5[th] and 95[th] percentile range (6.0 to 8.8 days) for OMI does not overlap at all with this percentile range in any of the other products. Other published estimates of the OMI-derived decay timescale after Kasatochi are also similarly fast (Krotkov et al. (2010) report a time of 8 to 9 days).

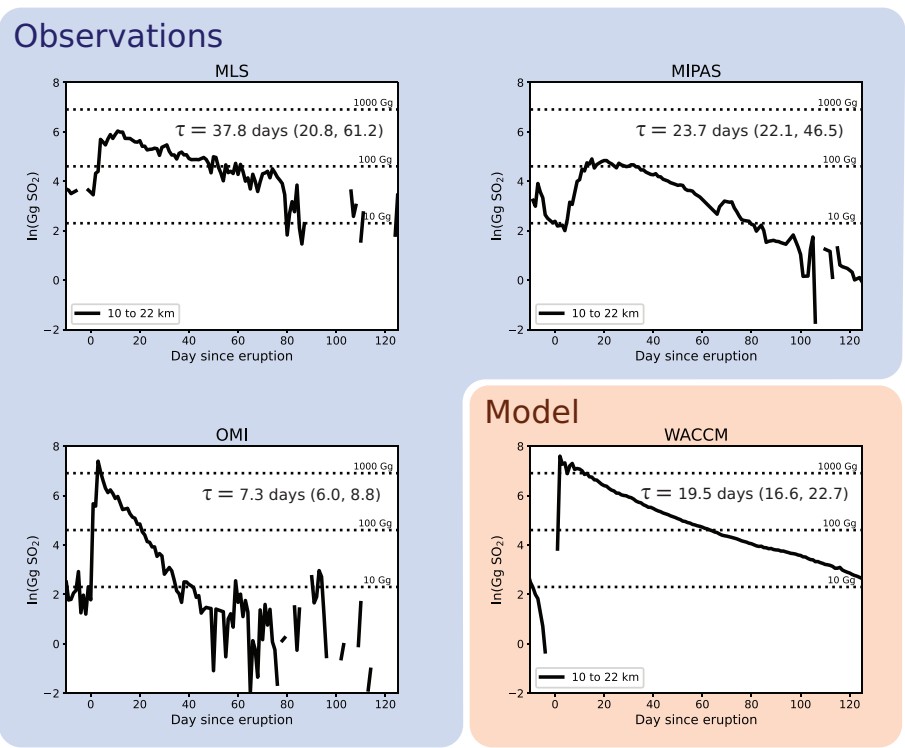

**Figure 6.** Comparison of stratospheric decay timescales for the 2008 Kasatochi eruption. Satellite products are highlighted in blue and the WACCM model in orange. For comparison with the total column stratospheric values from OMI, $SO_2$ masses for MLS, MIPAS, and WACCM, are calculated by vertically integrating from 10 to 22 km. For each product the median and $5^{th}$ and $95^{th}$ percentile (in parentheses) decay timescales are reported. Percentile ranges are derived using the method outlined in Sect. 2.4.

This difference in the OMI decay timescale also arises for the Sarychev and Nabro eruptions (Fig. B1 and Fig. B2, respectively). For both eruptions, the OMI value is much faster than either of the other three products, and there is nearly no overlap between the uncertainty ranges of OMI and the other products. There are a couple of possible explanations for this. While the OMI data used here is designed to give an estimate of $SO_2$ mass in the stratosphere (Sect. 2.1.3), there is potential for tropospheric $SO_2$ to influence this measurement. Tropospheric $SO_2$ will generally get removed much quicker than that in the stratosphere, and could be skewing the decay rates reported here. Additionally, there is a known bias in the OMI data due to the limited sensitivity of nadir instruments as the plume disperses (see Sect. 2.1). Both of these should be considered more carefully when analyzing OMI $SO_2$ following an eruption.

As an aside, we also comment on the differences between the model and MIPAS for the three eruptions. In the Kasatochi eruption, the discrepancy in the 10 to 22 km decay timescale between the model and MIPAS is much less than the other two eruptions (see Fig. 6, Fig. B1, Fig. B2). One plausible explanation for this is that the model puts a significant fraction of the

SO$_2$ from Kasatochi into the 14 to 18 km height bin, whereas for Sarychev and Nabro, there is relatively more mass in the 10
to 14 km height bin (Fig. S2). As OH decreases with height in the model, this would lead to slower (and thus closer to MIPAS)
decay times for the Kasatochi SO$_2$ when the larger vertical column is considered.

## 6 Estimating the stratospheric SO$_2$ burden

In addition to understanding the chemical fate of volcanically-emitted SO$_2$, quantifying its total stratospheric burden is key for
understanding the potential climate and chemical impacts of volcanic eruptions (Schmidt et al., 2018; Solomon et al., 1998).
Carn et al. (2016) provides a review of total SO$_2$ mass loading from volcanic eruptions occuring from 1978 through 2014.
These values are largely derived from nadir instruments (i.e. TOMS, OMI, IASI) and are often used as the standard reference
for volcanic SO$_2$ mass loading. Carn (2024) provides an updated data set of volcanic SO$_2$ injections into the stratosphere
through 2024, which we use as a benchmark for our estimates.

The eruptions analyzed in this work (Kasatochi 2008, Sarychev 2009, and Nabro 2011) feature a high plume height and high
VEI, which allowed them to inject upwards of 1 Tg into the stratosphere as reported by nadir sounders (Carn, 2024; Carboni
et al., 2016). Table 2 compares the SO$_2$ stratospheric mass loading for Kasatochi, Sarychev, and Nabro derived in this study
from satellite observations with previously published values.

We estimate the total stratopheric SO$_2$ burden after an eruption via two different methods. The first, referred to as "total
increase" in Table 2, uses the time series shown in Fig. 6, Fig. B1, and Fig. B2. For MLS and MIPAS, these are the time series
of SO$_2$ in the 10 to 22 km height bin. For each eruption's time series, we take the difference between each successive data point
(i.e., np.diff) and sum the positive values of the resulting array during the eruption period for each eruption. The eruption period
is based on volcanic activity reports provided by Venzke (2024). In the event that the maximum SO$_2$ mass occurs outside of
the eruption period (e.g. Kasatochi only erupted for two days, but peak SO$_2$ in the MIPAS data set occurs after this), we sum
the positive values of the differences until the day of maximum SO$_2$.

As discussed in more detail below, this method proves inadequate for accurate mass burden estimations. This is particularly
true for MIPAS, as the spectral bands measured by MIPAS saturate at high SO$_2$ concentrations. The inclusion of this method
in this paper provides a contrast compared to previously published total burden estimates derived from SO$_2$ decay timescale
estimates (e.g., Höpfner et al., 2015; Pumphrey et al., 2015, for MIPAS and MLS, respectively). This previous work estimates
the mass of SO$_2$ immediately after the eruption by fitting an exponential curve to the decay of SO$_2$. The mass at $t = 0$ is then
taken to be the total SO$_2$ emitted by the volcano. We use this same method (indicated "decay fit" in Table 2) where the window
used for the decay timescale calculation is varied as discussed previously. We report both a median mass loading and the 5[th]
and 95[th] percentiles to give a sense of the spread in the estimate.

**Table 2.** Estimated total SO$_2$ mass emitted into the stratosphere. For values calculated in this study, the utilized method is listed. Values from Höpfner et al. (2015) are indicated by H2015, and those from Pumphrey et al. (2015) are indicated by P2015. There was not a sufficient signal calculate the extrapolated SO$_2$ for Nabro using MLS. For values calculated in this paper, parentheses indicate the 5[th] and 95[th] percentiles.

### Kasatochi
#### August 7, 2008; (52.12°N, 175.51°W)

| Source | Method | Mass SO$_2$ (Gg) |
|---|---|---|
| **MIPAS (this study)** | total increase | 115 |
| **MIPAS (this study)** | decay fit | 382 (192, 444) |
| **MIPAS (H2015)** | | 899±154 |
| **MLS (this study)** | total increase | 440 |
| **MLS (this study)** | decay fit | 414 (311, 692) |
| **MLS (P2015)** | | 1350±38 |
| **OMI (this study)** | total increase | 1642 |
| **OMI (this study)** | decay fit | 1591 (767, 3329) |
| **Carn (2024)** | | 2000 |

### Sarychev
#### June 15, 2009; (48.01° N, 153.20° W)

| Source | Method | Mass SO$_2$ (Gg) |
|---|---|---|
| **MIPAS (this study)** | total increase | 210 |
| **MIPAS (this study)** | decay fit | 603 (422, 882) |
| **MIPAS (H2015)** | | 1473±299 |
| **MLS (this study)** | total increase | 221 |
| **MLS (this study)** | decay fit | 1137 (348, 2681) |
| **MLS (P2015)** | | 1160±180 |
| **OMI (this study)** | total increase | 232 |
| **OMI (this study)** | decay fit | 427 (336, 1243) |
| **Carn (2024)** | | 1200 |

### Nabro
#### June 13, 2011 (13.37° N, 41.70° E)

| | | Mass SO$_2$ (Gg) |
|---|---|---|
| **MIPAS (this study)** | total increase | 172 |
| **MIPAS (this study)** | decay fit | 363 (307, 424) |
| **MIPAS (H2015)** | | 539±117 |
| **MLS (this study)** | total increase | 812 |
| **MLS (this study)** | decay fit | — |
| **MLS (P2015)** | | 543±45 |
| **OMI (this study)** | total increase | 748 |
| **OMI (this study)** | decay fit | 498 (192, 1517) |
| **Carn (2024)** | | 1975 |

The masses calculated using the total increase method are generally much lower than those given by Carn (2024), and, with the exception of Sarychev, there is limited agreement between the observational products. While this approach avoids the uncertainty associated with using decay timescales to determine total mass burden, we don't find it adequate for an accurate determination of the total mass. This is particularly true of MIPAS due to its previously mentioned shortfall: it underestimates the total amount of $SO_2$ present at the start of the eruption when the plume is dense. Despite these shortcomings, this method does clearly illustrate how the different satellites capture the peak $SO_2$.

The total increase values reported here, with the exception of OMI during the Kasatochi eruption, are also not high enough to account for the observed aerosol loading and radiative forcing following the eruptions (Schmidt et al., 2018). However, despite the noise and interference issues, for Kasatochi and Nabro, MLS does get closer to the expected mass burden than MIPAS (though MLS still underestimates it significantly).

Previous estimates of total $SO_2$ using the decay fit method have generally led to good results for MIPAS and MLS when compared to other estimates (Höpfner et al., 2015; Pumphrey et al., 2015); however, the final value will be sensitive to how the exponential fit is calculated. As discussed previously, the details of this calculation can result in wide variability in the decay timescale (e.g. Fig. 6), and this directly translates to uncertainty in the mass loading. The $SO_2$ burdens calculated using the decay fit method in Table 2 highlight this. Moreover, a physical reason for this uncertainty could be the large range of decay timescales observed in the initial, dense volcanic plume (McKeen et al., 1984); a better understanding variations of the $SO_2$ oxidation rate within the lifetime of a given plume could help reduce this uncertainty.

We also note that the MIPAS and MLS $SO_2$ masses in some eruptions are significantly lower compared to the Carn (2024) values. Whether this reflects fractional stratospheric inputs or biases due to limitations of sampling by limb-sounding instruments would be a subject for future research. Additionally, our masses from MIPAS are lower than those reported in Höpfner et al. (2015). This is perhaps due to differences in the way the calculations are done. In Höpfner et al. (2015), masses are calculated (via an exponential fit) in $4\,km$ height layers and then summed to get the total stratospheric burden. Here we calculate the burden by calculating the masses from 10 to $22\,km$ and then applying the fit. Indeed, if we apply our method to the results shown in Figure 14 in Höpfner et al. (2015) (their plot of total stratospheric $SO_2$ burden after Sarychev), the resulting mass is comparable to what we report here.

Finally, we highlight that the uncertainty ranges for OMI are quite large, and this is due to the rapid decay timescales reported for OMI. Even though the spread among these timescales for a given eruption is low, the nature of the exponential fit means that the initial burden will be more sensitive changes in the decay timescale when the timescale is faster.

The main focus of the paper is on the decay times of the stratospheric $SO_2$ inputs from the indicated eruptions, and Table 2 explores the implications of such information for estimating the total stratospheric mass burden. Future work in this direction should consider the uncertainty in the decay timescale when fitting the curve with an exponential. Furthermore, our results indicate that simply summing the positive $SO_2$ perturbations from a volcano is not sufficient for getting an accurate mass burden in the data sets analyzed here.

## 7 Discussion and conclusion

Quantifying the total input and decay timescale of $SO_2$ is a key step in understanding the chemical fate of volcanically-emitted $SO_2$ in the stratosphere. In this work we utilize a combination of satellite products and a coupled chemistry-climate model to analyze the decay of the $SO_2$ perturbation from the three largest eruptions between 2004 and 2012: Kasatochi in 2008, Sarychev in 2009, and Nabro in 2011. This is the time period covered by all three of the satellite products used in this work. Smaller eruptions during this time period (e.g. Cordon Caulle, Grimsvötn, Redoubt) were not included due to the difficulty of detecting a clear decay of $SO_2$ in the MLS data set.

We compared the results between Michelson Interferometer for Passive Atmospheric Sounding (MIPAS) and the Mircowave Limb Sounder (MLS). Both limb-sounding instruments, the $SO_2$ retrievals from these products allow for a vertically-resolved analysis of the decay timescale. We report decay timescales in three different height bins (the same ones used by Höpfner et al. (2015)): 10 to 14 km, 14 to 18 km, and 18 to 22 km. In general, we find that uncertainty in the decay timescale is much larger in the MLS data set than in MIPAS. This is primarily a consequence of apparent interferences in the background seasonal cycle in the MLS data. While this seasonal cycle has been noted in previous work (Pumphrey et al., 2015) and is likely due to $HNO_3$ and $O_3$, it is not obvious what the form of the seasonal cycle should be, and this makes it inherently challenging to accurately determine the shape and magnitude of the volcanic perturbation. Furthermore, the MLS data is much noisier than the MIPAS data and features large negative values in the $SO_2$ mixing ratio. We suggest that the noise and negative bias is a consequence of pressure broadening; microwave emissions are more subject to pressure broadening, which obscures the signal received by the MLS instrument. This is further supported by the fact that the noise and negative bias is far less significant in the 18 to 22 km height bin (Fig. 3); the impacts of pressure broadening decrease with altitude (Pierrehumbert, 2010). Comparisons of the vertically-resolved decay timescales between satellite observations and a global climate chemistry model (WACCM, Gettelman et al. (2019)) indicate that the model generally predicts a faster decay timescale than the observations show, particularly at higher altitudes (Fig. 5). The 5[th] to 95[th] percentile range of decay timescales in the model also tends to be narrower than the observations. The reasons for these discrepancies require future investigation.

We also compare the the decay timescales of $SO_2$ within the whole stratospheric column. Here we include an additional observational data set from the Ozone Monitoring Instrument (OMI), which is a common choice for analyzing volcanic $SO_2$ decay (e.g., Carn et al., 2022; Zhu et al., 2020; Krotkov et al., 2010). We find that the timescale in the OMI data set is consistently the fastest across the three eruptions analyzed. The uncertainty range of the OMI data only minimally overlaps with the uncertainty ranges of any of the other three products for the Nabro eruption, suggesting that OMI might overestimate the rate of stratospheric $SO_2$ decay following an eruption. This is a bias, perhaps due to interference from tropospheric $SO_2$ and nadir instruments' limited sensitivity to dispersed plumes. It should be considered when analyzing volcanic $SO_2$ with OMI and other nadir-sounding instruments.

The range of decay timescales also limits the ability to accurately determine the initial mass loading following an eruption when using a constant decay timescale to extrapolate the $SO_2$ data back to the start of day of the eruption. Both Pumphrey et al. (2015) and Höpfner et al. (2015) utilize exponential fits to evaluate the initial $SO_2$ burden in MLS and MIPAS, respectively.

However, the significant variations in the decay timescale reported in this work translate to similar uncertainty in the total mass of $SO_2$ emitted by the volcano. This is a key quantity for understanding the climate and chemical impacts of volcanic eruptions, and this work suggests that constraining it using a exponential fit potentially comes with significant uncertainties.

  The high variability in decay timescales across observational products and the lack of consensus with the WACCM model makes it difficult to assess whether differences in decay timescale from one eruption to the next are due to different oxidation
processes or just the result of dynamics and the inherent difficulty of constraining and observing volcanic plumes. Zhu et al. (2020) argued that the relatively short decay timescale of approximately 7 days was indicative of heterogeneous oxidation on ash being an important process after the Kelut eruption in 2014. Kelut, like Nabro, was a tropical eruption but was known to inject large amounts of ash into the atmosphere that persisted for longer than usual (Vernier et al., 2016). The observations of Nabro in the 14 to 18 km height bin suggest longer $SO_2$ decay timescales ranging from about 10 to over 60 days, indicating
that the ash effect, if indeed significant, may well be specific to certain eruptions. Moreover, Zhu et al. (2020) use OMI data in their analysis, and as shown here, OMI results in fast decay timescales compared to MLS or MIPAS. Note that we don't present analysis of Kelut in this paper, as the MLS data for that eruption lacked a strong signal (not shown), and the MIPAS data ended in 2012, prior to the eruption of Kelut.

  The eruption of Hunga in 2022 has also been noted for its remarkably fast $SO_2$ $e$-folding time (Asher et al., 2023; Zhu
et al., 2022). A submarine volcano, Hunga injected an estimated 150 Tg water vapor ($H_2O$) and $0.41 \pm 0.01$ Tg $SO_2$ into the stratosphere, and the rapid decay of $SO_2$ has been attributed to a significant increase in OH following the $H_2O$ injection (Asher et al., 2023). The plumes from the eruptions also reached over 30 km above the surface (Asher et al., 2023). Given the unusual nature of the eruption, we don't include a quantitative analysis of it here. Rather, we mention it as a further example of the possible variability in the conditions influencing volcanic $SO_2$ oxidation.

Our work suggests that the current $SO_2$ data reported by available observational products are subject to significant uncertainty when examining the stratospheric decay of volcanic $SO_2$. The varying strengths and shortcomings of the different observational products should be accounted for when using them to determine chemical mechanisms and $SO_2$ mass loading. Furthermore, the forthcoming loss of MLS (the only limb-sounding $SO_2$ instrument in operation with continuous coverage over global latitudes and longitudes) will leave a significant gap in our ability to monitor the stratosphere.

*Code and data availability.* All satellite data used in this study are publicly available. MIPAS (with registration): https://www.imk-asf. kit.edu/english/308.php. MLS: https://disc.gsfc.nasa.gov/datasets/ML2SO2_005/summary?keywords=so2. OMI: https://disc.gsfc.nasa.gov/ datasets/OMSO2_003/summary. CESM2-WACCM6 data, along with scripts for analysis and generating figures are provided upon request.

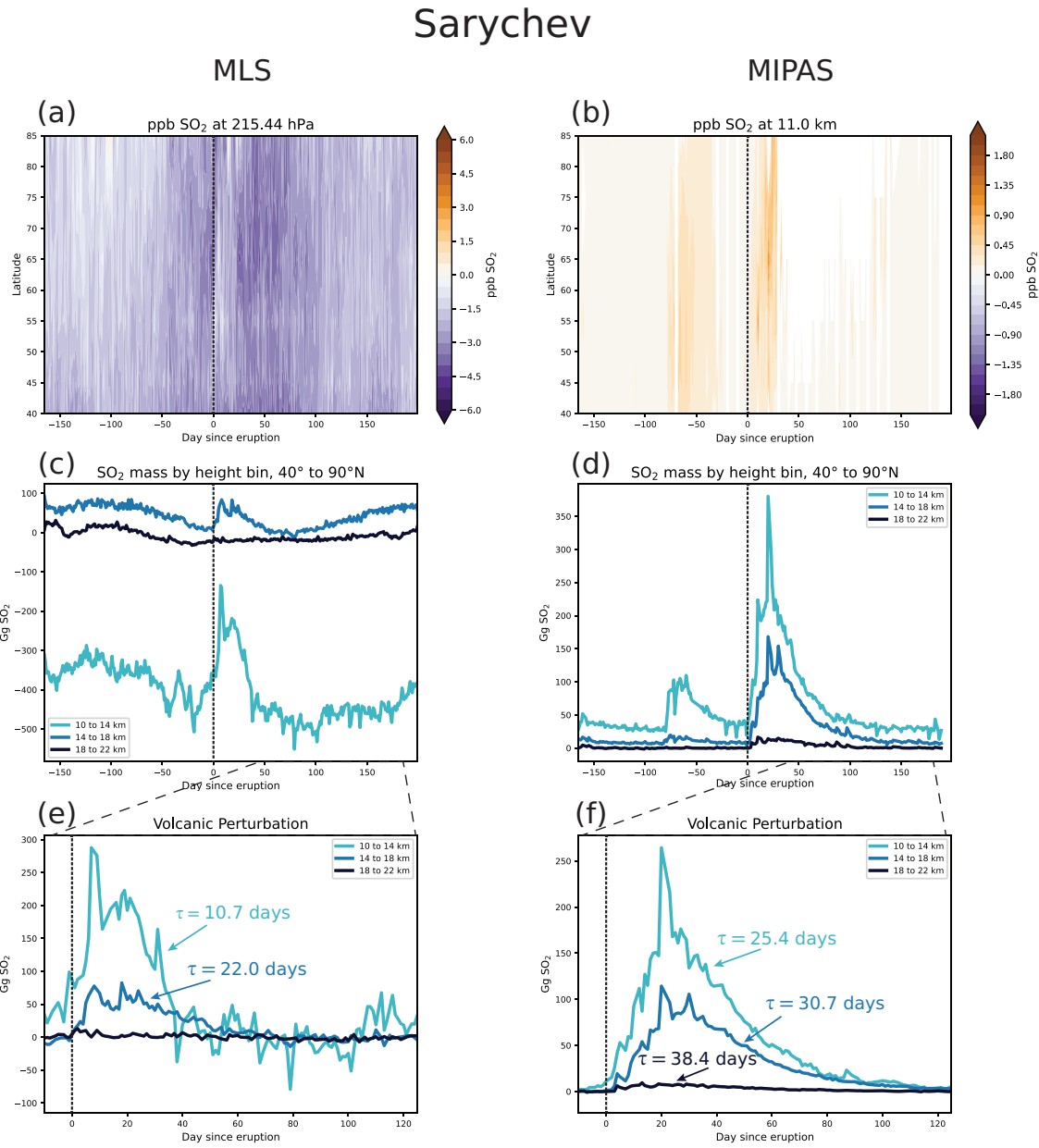

**Figure A1.** Comparison of MLS and MIPAS observations during 2009 and the Sarychev eruption, using the same format as Fig. 3. Day 0 is June 15, 2009. Note that the smaller signal seen in the MIPAS data prior the the Sarychev eruption is due to the eruption of Redoubt in March, 2009 (Carn, 2024).

# Nabro

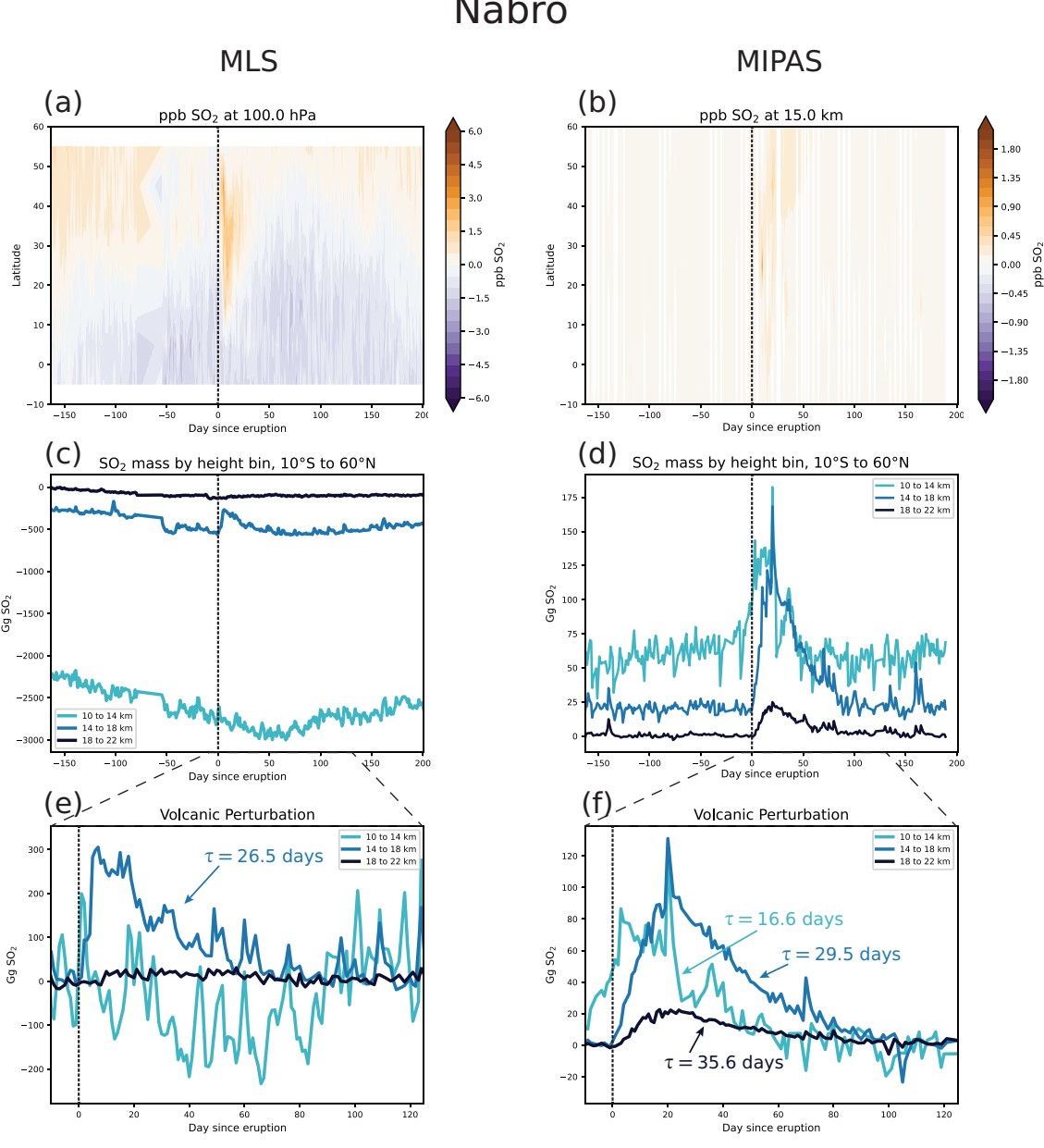

**Figure A2.** Comparison of MLS and MIPAS observations during 2011 and the Nabro eruption, using the same format as Fig. 3. Day 0 is June 13, 2011.

## Appendix B: Comparison with OMI

# Sarychev

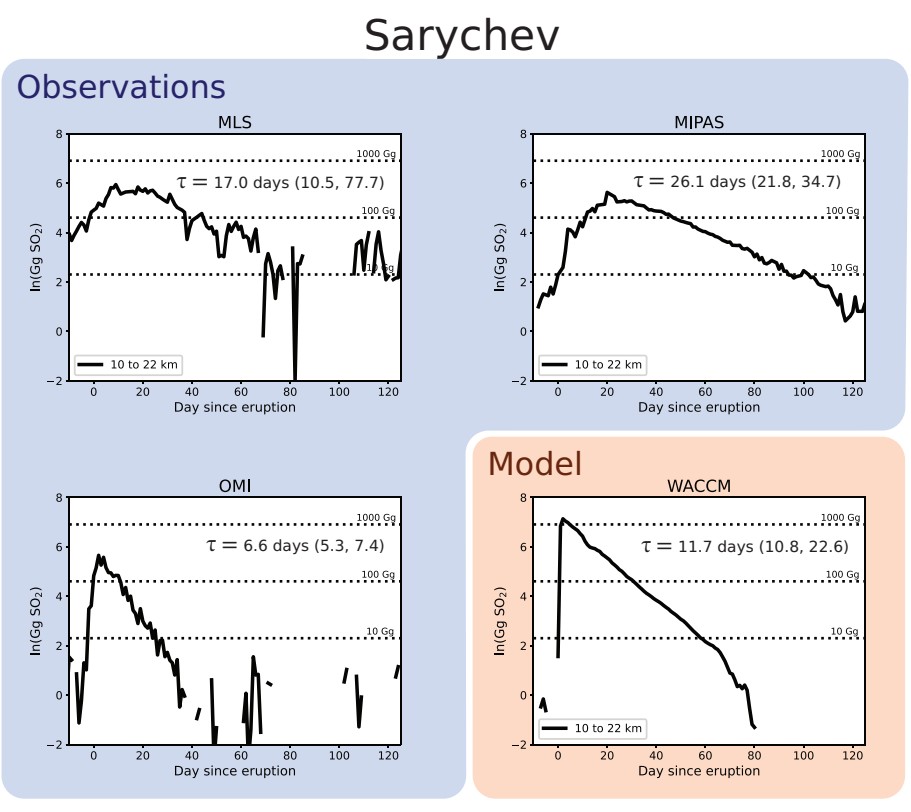

**Figure B1.** Comparison of stratospheric decay timescales for the 2009 Sarychev eruption, using the same format as Fig. 6.

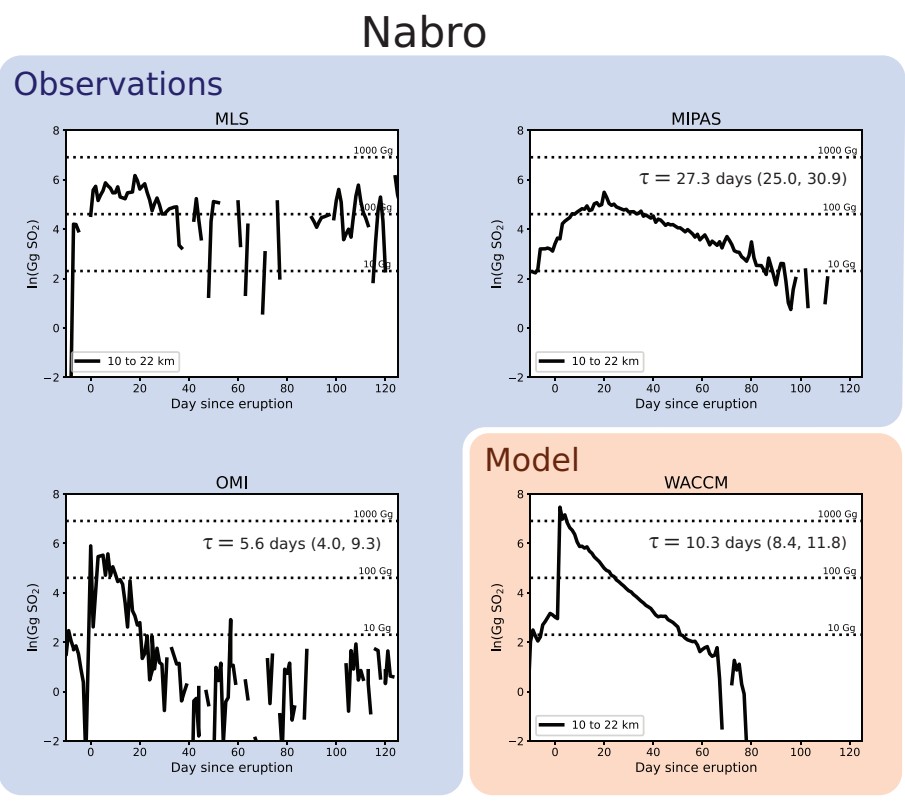

**Figure B2.** Comparison of stratospheric SO$_2$ decay timescales for the 2011 Nabro eruption using the same format as Fig. 6. Note that there was not a sufficient signal in the MLS data to report a timescale.

*Author contributions.* P.A.N. and S.S. developed the study, and P.A.N. conducted the analysis with input from K.S. and S.S. All authors contributed to the writing and revision of the manuscript.

*Competing interests.* The authors declare that they have no conflict of interest.

*Acknowledgements.* P.A.N. acknowledges support from the Presidential Graduate Fellowship at MIT and the National Science Foundation Graduate Research Fellowship under Grant No. 2141064. S.C. acknowledges support from the NASA MEaSUREs program (Grant No. 80NSSC24K0922). S.S. appreciates support by NSF grant 2316980. The authors would like to thank Hugh Pumphrey for helpful conversations and insights during the research process.

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
