# Peer review of "Quantifying the decay timescale of volcanic sulfur dioxide in the stratosphere"

_EGUsphere, 2024_

## Author Comment (AC2)

**Quantifying the decay rate of volcanic sulfur dioxide in the stratosphere**

Response to reviewers in red
* * *
**Reviewer #1**

*l.22: '…is produced naturally in seawater':*

This is only one of the several sources of OCS. It is even not clear if the major source is direct emission from sea-water, or if OCS is mainly produced by oxidation of CS2 and DMS (which originate in sea-water) (e.g. Kremser et al., 2016). Please be either more specific (or more general) here.

We have updated the language to be more specific as follows (see lines 22-25 in the revised manuscript):

An important source of stratospheric $SO_2$ in non-volcanic conditions is the photolysis of carbonyl sulfide (COS), which is the most abundant sulfur-containing gas in the atmosphere (Kremser et al., 2016). Important sources of COS include its direct flux from the ocean, oxidation of marine-originating dimethyl sulfide and carbon disulfide, and direct and indirect anthropogenic emissions, among others (Kremser et al., 2016).

*l.81: '…they potentially provide greater sensitivity to volcanic SO2'; l.86: 'the greater sensitivity of limb sounders may be advantageous'; l.332: 'This is likely to be a bias in the OMI data, perhaps due to the limited sensitivity of nadir instruments as the plume disperses…'; l.399: 'This may be a bias…'*

I acknowledge the scientific caution when attributing biases in stratospheric $SO_2$ decay timescales to nadir-viewing instruments. However, the evidence presented in this study, as well as in previous works, strongly supports the existence of such a bias. This bias arises from the limited sensitivity of nadir-viewing instruments to diluted stratospheric $SO_2$ amounts compared to limb-viewing instruments, which benefit from significantly longer optical path lengths (several hundred times greater) through the dispersed plume. Given the robustness of this evidence, I recommend adopting clearer language to describe this phenomenon. Such clarity is important to avoid potential misinterpretation of nadir-viewing data, particularly in the future, when a substantial

volume of nadir-derived data remains available while limb-measurements might no longer be conducted.

This is a great point, and we have adjusted the language throughout the manuscript for increased clarity. For example, the text:

 "This is likely to be a bias in the OMI data, perhaps due to the limited sensitivity of nadir instruments as the plume disperses"

has been changed to:

 "Additionally, there is a known bias in the OMI data due to the limited sensitivity of nadir instruments as the plume disperses" (see line 397-398 in the revised manuscript).

*l.98: 'high spectral resolution'*

Should this not read 'high spatial resolution'?

 When downloading the data from https://imk-asf-mipas.imk.kit.edu/mipas/, we had to select either "V5 high spectral resolution" or "V5 low spectral resolution". We used the V5 high spectral resolution data.

*l.106: 'are reported on pressure levels with an approximate spacing of…'*

The reported retrieval-grid of remote sensing data is generally not  equal to its spatial resolution. Therefore, I would suggest to add the information on resolution from here: https://mls.jpl.nasa.gov/data/v5-0_data_quality_document.pdf, p. 157

Thank you for the suggestion. We updated the resolution numbers and added a citation for the data quality document. See lines 134-136 in the revised manuscript.

*l.146: 'to the pressure coordinate of the MIPAS data'*

Shouldn't this read 'to the altitude coordinate…'?

Yes it should! Thank you for the catch. We changed the text as suggested.

*l.236: 'The reason for this remains unclear, and no explanation or documentation for this difference was found in the literature.'*

You might try to contact the MLS-team for a possible explanation(?)

We have had personal correspondence with Hugh Pumphrey, one of the leading scientists working with MLS. He was also perplexed by the difference, and we did not get a resolution talking with him.

**Chapter 3.2: 'Background seasonal cycle in the MLS data'**

Is there any possibility to infer the disturbing seasonal cycle from comparing different years with less volcanic influence?

[Figure]

Thank you for the question and suggestion. The spread from year-to-year is larger than the intra-annual variability. In our method we sampled just the variability within a given year, and this still led to uncertainty. Given this plot here, which shows the seasonal cycle with large volcanic signals omitted, we don't think using non-volcanic years will increase confidence.

***l.290, Table 1:***

I strongly recommend consolidating all the information from Table 1 and Table B1 into a single comprehensive table. Additionally, I suggest including the uncertainties reported in Höpfner et al. (2015) and the results from the WACCM simulations, including those for the 18–22 km layer if available. This enhancement would greatly improve the readability of the discussion in Chapter 3.3, allowing readers to follow the analysis more easily without needing to consult multiple tables.

Thank you for the suggestion. We agree this improves readability, and Table 1 and the surrounding discussion in section 3.3 has been updated accordingly.

***l.290: 'Their values show a clear increase of e-folding time with height, which is not as apparent in our results.'***

However, the decay timescales provided here also show increasing values (with one exception of MLS in case of Kasatochi). Further, I suggest also to try to provide best-estimates from your analysis of the decay-timescales at 18-22 km for MIPAS in case of Kasatochi and Sarychev – when looking at Fig. 2 and Fig. A1, there seems to be a clear signal.

We adjusted the text following Table 1 to read: "Their values show a clear increase of decay rate with height, which is generally also seen in our results (with the exception of some of the values for Kasatochi)" (lines 346 to 347 in the revised manuscript).

We also added MIPAS estimates for Kasatochi and Sarychev in the 18-22 km height bin in Fig. 2, Fig. 4. Fig. A1, and Table 1.

***l.295-314: comparison to WACCM***

The discussion may give the impression that the uncertainties in the decay timescales derived from the measurements are too large to allow for meaningful comparisons with the model. However, upon examining Figures 5, C1, and C2, it appears that for the Sarychev and Nabro eruptions, the model significantly underestimates the decay timescales compared to the limb-sounding datasets, whereas this discrepancy is less pronounced for the Kasatochi eruption. Could you provide possible explanations for this observed difference?

One possible explanation for the observed difference could be the fact that the model puts most of the $SO_2$ from the Kasatochi eruption into the 14-18 km height bin, whereas most of the $SO_2$ for the Sarychev eruption is in the 10-14 km height bin. The model also puts a substantial amount of $SO_2$ in the 10-14 km height bin for the Nabro eruption. As OH decreases with height in the model, $SO_2$ is going to get oxidized and removed more slowly the higher up it is. This could account for the slower decay timescales (and thus closer to MIPAS) reported for Kasatochi (as compared to Sarychev and Nabro) when looking at the larger vertical column. We added a figure (S2) to the supplement showing this and a brief discussion in lines 400-405 in the revised manuscript.

[Figure]

***l.355: 'our main focus here is on the decay times of the stratospheric inputs of the indicated eruptions and not the total stratospheric mass.'***

The entire chapter 5 of the manuscript (as well as section 2.5 'Calculation of total stratospheric SO2 burden') is dedicated to the 'Estimating the stratospheric SO2 burden'. Therefore, I don't understand this statement. I would suggest to delete this sentence and extend chapter 5 a bit by extending Table 2 to include the estimations of stratospheric SO2 mass by the extrapolation methods used by Pumphrey et al. (2015) and Höpfner et al. (2015). I would also suggest to add the results (M(t0)) from the fits performed in the present work. It should be made clear that the method described here in section 2.5 is not adequate to calculate the total stratospheric SO2 burden in case of MIPAS and, to a less extend, also for MLS.

We have adjusted the sentence you highlighted to read:

"...our main focus here is on the decay times of the stratospheric inputs from the indicated eruptions, and we comment on the implications of such information for

determining the total stratospheric mass burden" (lines 421-423 in the revised manuscript).

Your comment about presenting the inadequacy of method presented in Section 2.5 is well taken, and we have adjusted the text in Section 2.5 and Section 5 to emphasize this more clearly. For example, see lines 227-230 in the revised manuscript.

Finally, we added values from Höpfner et al., (2015) and Pumphrey et al., (2015) to Table 2. With respect to your comment about including our estimates of M(t0), we think that including these wouldn't assist this section of the paper. Our goal is to show how a different way of calculating $SO_2$ impacts the results, and previous work has already given estimates on total $SO_2$ by fitting M(t0). We feel including these values will also clutter up this section of the paper and the table.

*l.400: '…and should be considered when analyzing volcanic SO2 with OMI'*

I would suggest to add here: 'and other nadir-sounding instruments'.

 Thank you for the suggestion; this text was added.

*l.413-417: eruptions with ash*

The eruption of Puyehue in June 2011 was also rich in ash. Have you tried to inspect that one for any effects on SO2 lifetime? (e.g. Griessbach et al., 2016, doi:10.5194/amt-9-4399-2016)

Thank you for the comment, as it is a good point. We considered other eruptions during the time period of overlap between MLS and MIPAS. However, we focus here on three specific eruptions in this paper because they were large enough to allow for a calculation of the e-folding time of $SO_2$.  Puyehue, as well as other notable eruptions during this time period such as Grimsvotn had too weak and noisy of a signal, particularly in the MLS data, to calculate the decay rate. As we are interested in comparing MLS and MIPAS, we leave these smaller eruptions out of the analysis.

*l.424: 'Our work suggests that the current SO2 data reported by available observational products are subject to significant uncertainty when examining the stratospheric lifetime of volcanic SO2 and suggests that more precise data is needed if chemical mechanisms and SO2 mass loading following an eruption are to be elucidated using observed decay times.'*

On one hand, I support this statement, particularly considering the imminent loss of limb-sounding capabilities for stratospheric $SO_2$ observations, which will create a significant gap in our ability to monitor the stratosphere. On the other hand, I find the statement somewhat overly general. As noted in the manuscript, each observational technique has specific advantages and limitations in quantifying stratospheric $SO_2$. Therefore, to effectively evaluate and refine models, it may be more appropriate to tailor comparisons to align with the strengths of each dataset. For example, model results could be compared directly with nadir and MLS data closer to the eruption time, while comparisons with IR limb-sounding datasets might be more suitable for periods several weeks after the eruption.

Thank you for the suggestion. We have modified the last paragraph (lines 494-495 in the revised manuscript) to read as follows:

"Our work suggests that the current $SO_2$ data reported by available observational products are subject to significant uncertainty when examining the stratospheric decay of volcanic $SO_2$. The varying strengths and shortcomings of the different observational products should be accounted for when using them to determine chemical mechanisms and $SO_2$ mass loading. Furthermore, the forthcoming loss of MLS (the only limb-sounding $SO_2$ instrument in operation) will leave a significant gap in our ability to monitor the stratosphere."

---

## Author Comment (AC3)

**Quantifying the decay rate of volcanic sulfur dioxide in the stratosphere**

Response to reviewers in red
* * *
**Reviewer #2**

I was hoping to see a 'back of the envelope' check against the total aerosol eruption mass burden – the end product of SO2 oxidation. This would be a useful addition. (See **Schulte et al., 2023,https://doi.org/10.5194/AMT-680 16-3531-2023 on computing the total mass).** There are a number of stratospheric aerosol sources you can use – but probably GLOSSAC is the best. This sort of 'stupidity check' would confirm that the SO2 estimates agree with aerosol production - which is why we care about this.

Thank you for the comment and the suggestion. We have added a discussion about this point to the introduction (lines 49 to 51 in the revised manuscript), as well as a supplementary figure, which is copied below (Fig S1.)

[Figure]

In regards to your suggestion about GLOSSAC, looking through the various GLOSSAC data products available, we only found data products available on a monthly resolution (https://asdc.larc.nasa.gov/project/GloSSAC). As we are interested in processes happening on the order of weeks, monthly resolution data won't have the necessary level of detail. However, using sulfate aerosol burden estimates from MIPAS (see Günther et al., 2018; https://doi.org/10.5194/acp-18-1217-2018), we did a rough estimation for the timescale of sulfate aerosol formation for the 2009 Sarychev eruption. This is using MIPAS data from 10.5 to 22.5 km. The ~28 day time scale of sulfate aerosol formation aligns well with our estimate of a $SO_2$ decay time scale for the eruption (25-30 days).

An additional point we'd like to make here is that detecting perturbations to the stratospheric aerosol layer is not trivial due to the constantly varying background (e.g., Solomon et al., 2011; DOI: 10.1126/science.1206027). This makes defining an appropriate baseline for calculating timescales associated with a perturbation challenging (and is a similar issue to that seen with MLS in our analysis.)

The MLS algorithm, as I understand it, generates negative mixing ratios for data on the edge of observability. The correct way to deal with these are to average the data over larger regions including both positive and negative mixing ratios. I was looking for a discussion of this and mention of MLS validation also found in Livesey et al. (2022) (found at https://mls.jpl.nasa.gov/eos-aura-mls/documentation.php). Discussion of how to use the data including quality flag screening that is appropriate for SO2 is also found there and should be mentioned in the data description (lines 65-75). An equivalent MIPAS discussion is needed.

We added the following text to expand on this detail of the MLS dataset (lines 140 to 144 in the revised manuscript):

> "The MLS documentation highlights that the retreival algorithm can generate negative mixing ratios, and the correct way to deal with these is to average over a sufficiently large horizontal area (Livesay et al., 2022). We apply all of the suggested masking for the data given in (Livesay et al., 2022), and we average our data over 10° latitude bands. Even after masking and averaging, negative mixing ratios are prominent in the MLS data, particularly lower in the atmosphere."

We also added details for how we did the data masking based on the suggested guidelines for MIPAS (lines 128 to 129 in the revised manuscript):

> "As suggested within the MIPAS data files, we select valid data by only using points where visibility == 1 and akm_diagonal > 0.03."

The division of the SO2 into three separate regions (10-14), (14-18), (18-22) made me uncomfortable. At high latitudes in winter these regions are all in the stratosphere – in the summer the 10-14km may include the troposphere. In the tropics (as with Nabro and many other eruptions), only 18-22km is entirely in the stratosphere. This distinction can play an important part since the water vapor content and OH concentration (eq. R1)of these layers can be quite different – upper troposphere vs lower stratosphere – and thus will affect the decay rate. Since MLS and MIPAS also make water vapor measurements, the water vapor content can be added to the analysis. It seems to me that the the authors should have used two layers - below the tropopause and above the tropopause - rather than what was done here. It is easy to get tropopause height information from reanalysis data sets (GFS, MERRA2, ERA5).

Thank you for the comment. The main rational behind using the three layers is that it allows for comparison to previous work by Höpfner et al., (2015). We agree, however, that whether or not these layers fall in the stratosphere is going to be highly latitude dependent. As such, we have included a more detailed discussion of where these layers fall as a function of time-of-year and latitude in the beginning of Section 3.3 (lines 314 to 320 in the revised manuscript):

> "In particular, the tropopause in the tropics during the local summer is around 16 km, whereas that for the high northern hemisphere latitudes is closer to 11km (Hoffmann and Spang, 2022). As such, the majority of the three layers considered in this analysis are likely to be in the stratosphere for the Kasatochi and Sarychev eruptions. After the Nabro eruption, likely only the 18 to 22km layer was initially fully in the stratosphere; however, the plume was quickly advected to higher latitudes—where the tropopause is lower—by the Asian Monsoon anticyclone in just a few days (Clarisse et al., 2014)."

We think that in order to facilitate comparison with past work, keeping the analysis focused on these three layers is the best choice. However, we have added a comment on the validity of this choice in light of the concerns you raise. See the updated text in Section 2.3 (lines 185 to 188 in the revised manuscript):

> "Given the variation in tropopause height with latitude, the 10 to 14km and 14 to 18km layers won't necessarily be entirely in the stratosphere in low latitudes (Hoffmann and Spang, 2022). However, we use the vertical divisions here for consistency with past work, and future work could consider a division based on tropopause height."

The authors neglect the transport between the layers. Exchange between layers needs to be discussed as possibly influencing the decay rate.

We had considered the potential role of transport in influencing our decay rate but came to the conclusion that vertical transport is too slow to significantly affect the decay rate. The processes of interest here are on the order of weeks, and at most we are using a window of 25 days in our calculations. In comparison, typical time scales associated with vertical transport in the stratosphere are on the order of tenths of a milimeter per second or hundreths of a kilometer per day (e.g., Butchart (2014); 10.1002/2013RG000448).

There is evidence in self-lofting of volcanic plumes, similar to that seen in wildfire plumes (e.g., Khaykin et al., 2022; https://doi.org/10.1038/s41598-022-27021-0). However, this has not been noted for the volcanoes analyzed here. Nonetheless, it is a potential source of uncertainty.

Both of these points have been elaborated on in the second paragraph of Section 3.3; see lines 323 to 336 in the revised manuscript.

The authors are using a zonal mean SO2 on a 10° latitude grid (ln 153). It seems like they could also construct a tighter latitude grid (say 5°) and a longitude grid and select high SO2 regions which might reduce the uncertainty (e.g. Fig. 2). I would like to see how this affects their decay rate and agreement between the two satellite instruments.

In general, we find that using smaller latitude bands has almost no impact on the shape of the perturbation and the decay. We have included an example from the 2009 Sarychev eruption below. On the left is the time series of the MIPAS $SO_2$ perturbation calculated using 10° latitude bands, and on the right is that using 5° latitude bands. The two curves are nearly identical.

Left: 10° bands                                        Right: 5° bands

[Figure]

[Figure]

I don't think adding OMI SO2 helps the paper at all. In fact, it just adds noise, not signal. This is because the OMI total column includes massive amounts of tropospheric SO2 (for most eruptions) which – as the authors note – probably explains the significant differences in total SO2 mass and decay rate. If you add OMI you might as well take a look at SO2 measurements from AIRS (mentioned line 71) as well for completeness. Anyway, I suggest you just drop this section – it really adds nothing.

Thank for the comment. We are in favor of keeping the OMI section in the paper as we think that the comparison between MLS and MIPAS (both limb-sounding instruments) and OMI (a nadir-sounding instrument) helps identify some of the important discrepancies that arise between the two, particularly in light of some studies that have used OMI-based measurements as evidence for different $SO_2$ oxidation pathways (e.g., Zhu et al., (2020)). As you mention, there are other nadir-sounders we could have used, and indeed, a comparison between these and the limb-sounders analyzed here could be a useful thing to do in the future. For now, we think such a step is beyond the scope of the current paper. We adjusted language in lines 375 to 379 in the revised manuscript to better emphasize this:

"We compare the results from MLS, MIPAS, and WACCM with $SO_2$ retrievals from the Ozone Monitoring Instrument (OMI). OMI is a popular choice in recent work examining the decay of $SO_2$ following eruptions (e.g., Carn et al., 2022; Zhu et al., 2020; Krotkov et al., 2010), and we include an analysis of it here for a comparison of how limb-sounders and nadir-sounders capture the removal of volcanic $SO_2$. Note that OMI is just one of several nadir-sounding instruments that measure $SO_2$, and a detailed comparison with other instruments (such as AIRS) is left to future work."

Furthermore, one of the issues with OMI is it likely does contain tropospheric influence. This, combined with its known detection limit issues, are likely to contribute to a faster-than-realistic decay of $SO_2$, which is what we try to show by including it in this paper. We have updated the language in Section 4 to highlight this point (lines 393 to 398 in the revised manuscript):

"While the OMI data used here is designed to give an estimate of $SO_2$ mass in the stratosphere (Sect. 2.1.3), there is potential for tropospheric $SO_2$ to influence this measurement. Tropospheric $SO_2$ will get removed much quicker than that in the stratosphere, and could be skewing the decay rates reported here. Additionally, there is a known bias in the OMI data due to the limited sensitivity of nadir instruments as the plume disperses (see Sect 2.1). Both of these should be considered more carefully when analyzing OMI $SO_2$ following an eruption."

The WACCM simulation is interesting but barely discussed (Fig. 5). Take a look at the water vapor in WACCM. Was it the same as MLS observations? This might explain the accelerated decrease.

This is an interesting suggestion, as water vapor differences could very well be the culprit here. However, previous work comparing MLS water vapor to WACCM shows that there is generally very good agreement between the two in the region of interest for this paper. For example, see Figure 3 from Froidevaux et al., (2019); (https://doi.org/10.5194/acp-19-4783-2019):

[Figure]

We added a brief discussion about this in the second-to-last paragraph of section 3.3 (lines 358-360 in the revised manuscript):

> "One reasonable issue could be differences in water vapor and between the model and observations. However, comparisons between WACCM and MLS water vaper generally show strong agreement, and it is not clear that this should be the main culprit (Froidevaux et al., 2019)."

**Minor comments:**

Add layer labels to Fig. 2

We think the legend sufficiently conveys which lines correspond to the different layers, and adding labels directly on the plot would add unnecessary clutter to the figure.

Line 287 'less uncertain'  - how about 'better'

Thank you for the suggestion. We updated the text.

Fig 4.  Why not connect the dots vertically. The figure – as is, is a little hard to read.

Our updated figure 4 is shown here and has been added to the paper.

[Figure]

Line 418 "Honga-Tonga"  - the APARC group recommends using 'Honga' not HTHH or HT or other acronyms.  The Honga eruption is a good example where hydrolysis probably played a critical role in accelerating the decay of SO2 and conversion to aerosols as noted.  This is why I recommend the authors also take a look at H2O in other regions.

We changed the wording as suggested.

---

## Author Comment (AC6)

**Quantifying the decay rate of volcanic sulfur dioxide in the stratosphere**

**Response to editor in red**

Note: An additional figure and section were added, and figure and section numbers in our responses refer to those in the updated manuscript.
* * *
L50: This introduction of a result of this study is misplaced in the introduction. Please move to the results section.

We have moved this result to a new section (Sect 3, lines 220-226) in the results part of the paper. Additionally, we moved the relevant figure from the supplement to the main paper.

L62: "Do not tell the whole story" implies there is something new we will learn from MIPAS, but the sentence goes on to give a result which is consistent with the column measurements discussed in the prior sentence. Some different language would make this easier to understand.

Thank you for the suggestion. We have changed the sentence to:

> "However, total column measurements obscure vertical variations in $SO_2$ oxidation rates within the plume of a particular eruption: Hopfner et al., (2015) use vertically resolved observations from the Michelson Interferometer for Passive Atmospheric…" (lines 58-60)

- L84: "concentration" is a physical quantity that is not used much in this study, I wonder if mass would be a better example to use here?

Thank you for the comment. We have changed the word to "mass" in Section 1.1.

- L86f: I am fairly certain there is one too many derivatives here, and a missing negative sign. What you plot later and take the slope of is just ln(W(t)).

Thank you for the catch! We removed the extra derivative.

- L227, 229: Colloquially, we often talk about the impact of volcanos, etc., but formally it's really the "eruption" that is important here, not the volcano.

We adjusted the wording as requested.

- L251: "elevation" should be "altitude"

We have adjusted the wording as suggested here and throughout the manuscript.

- One of the main conclusions is that uncertainty in the decay timescale "limits the ability to accurately determine the initial mass loading following an eruption when fitting applying an exponential decay to the SO2 data". But you do not calculate the resulting uncertainty in the SO2 estimate. This is a clear hole in the argument that could be easily filled, since for every value of decay timescale, there will be a corresponding value of the injection amount. Instead, you provide a different type of injection estimate, based on the positive increases in the SO2 timeseries. But I find this estimate unconvincing, since it is clear that due to instrument and/or sampling issues, the timeseries are biased low in the first days when the timeseries apparently increases, but it is likely that SO2 is being removed through that period. Plus, since sampling can lead to random errors in the total mass, by discounting negative tendencies in the timeseries it could theoretically be possible that you bias the result. In any case, estimating an injected SO2 amount is such a natural part of assuming the exponential model of SO2 decay that it would fit well here and provide the basis of your conclusions about the method.

This is a good point, and we have reworked Section 6 and updated Table 2 to include this estimate of the total SO2 burden. Please see the revised document for the changes.

- In your response to referee comment "Is there any possibility to infer the disturbing seasonal cycle from comparing different years with less volcanic influence?" I don't understand your reply that "The spread from year-to-year is larger than the intra-annual variability." It appears from your plot that the intra-annual spread is around 500 Gg, and the spread between years is around 100-200 Gg. It appears that the referee's idea could be useful and could decrease the uncertainty in the fits, given that the spread in non-volcanic years appears potentially smaller than the spread in the example shown in Fig. 3.

Thank for you bringing this up. Our point could have been stated more clearly in our initial reply. We are trying to emphasize that at each day of the year, there is a spread of around 100-200 Gg. Supposing we take the mean of the background years as the reference seasonal cycle, the actual disturbing seasonal cycle could be $\pm$50-100 Gg

offset from the reference. This would impart a large bias on the calculated decay rate, thus limiting our ability to constrain the uncertainty.

The interfering background seasonal cycle is interesting in its own right, and to our knowledge under-explored in the literature. Given that it is likely due to $O_3$ and $HNO_3$, perhaps one might be able to calibrate the seasonal cycle based on observed values of these trace gases. However, we feel this would be beyond the scope of the current study. Nonetheless, we have added further discussion on this point (lines 305-312) in Section 4.2 as well as a supplementary figure (S2).

> "The approach outlined above samples possible seasonal cycles using the observed time series in a given year of an eruption. One could also potentially infer the interfering seasonal cycle from years without large volcanic eruptions; however, this presents its own challenges. There is a 100 to 200 Gg spread in $SO_2$ mass on any given day of the year in non-volcanic conditions (Fig. S2). Using the mean across years for the background seasonal cycle could result in an offset of $\pm 50$ to 100 Gg from actual disturbing seasonal cycle. This difference is large enough to impart a significant bias in the estimated decay rate, thus limiting our ability to constrain the uncertainty. The seasonal cycle and inter-seasonal variability in MLS $SO_2$ is interesting in its own right: it warrants further investigation and could potentially be calibrated based on the observed amounts of $O_3$ and $HNO_3$ but this is beyond the scope of the current paper."

---

## Author Comment (AC7)

**Quantifying the decay rate of volcanic sulfur dioxide in the stratosphere**

Response to reviewer in red
Note: An additional figure and section were added, and figure and section numbers in our responses refer to those in the updated manuscript.
* * *
**Reviewer #1**

*l.22: '…is produced naturally in seawater':*

This is only one of the several sources of OCS. It is even not clear if the major source is direct emission from sea-water, or if OCS is mainly produced by oxidation of CS2 and DMS (which originate in sea-water) (e.g. Kremser et al., 2016). Please be either more specific (or more general) here.

We have updated the language to be more specific as follows (see lines 22-25 in the revised manuscript):

> "An important source of stratospheric $SO_2$ in non-volcanic conditions is the photolysis of carbonyl sulfide (COS), which is the most abundant sulfur-containing gas in the atmosphere (Kremser et al., 2016). Important sources of COS include its direct flux from the ocean, oxidation of marine-originating dimethyl sulfide and carbon disulfide, and direct and indirect anthropogenic emissions, among others (Kremser et al., 2016)."

*l.81: '…they potentially provide greater sensitivity to volcanic SO2'; l.86: 'the greater sensitivity of limb sounders may be advantageous'; l.332: 'This is likely to be a bias in the OMI data, perhaps due to the limited sensitivity of nadir instruments as the plume disperses…'; l.399: 'This may be a bias…'*

I acknowledge the scientific caution when attributing biases in stratospheric $SO_2$ decay timescales to nadir-viewing instruments. However, the evidence presented in this study, as well as in previous works, strongly supports the existence of such a bias. This bias arises from the limited sensitivity of nadir-viewing instruments to diluted stratospheric $SO_2$ amounts compared to limb-viewing instruments, which benefit from significantly longer optical path lengths (several hundred times greater) through the dispersed plume. Given the robustness of this evidence, I recommend adopting clearer

language to describe this phenomenon. Such clarity is important to avoid potential misinterpretation of nadir-viewing data, particularly in the future, when a substantial volume of nadir-derived data remains available while limb-measurements might no longer be conducted.

This is a great point, and we have adjusted the language throughout the manuscript for increased clarity. For example, the text:

 "This is likely to be a bias in the OMI data, perhaps due to the limited sensitivity of nadir instruments as the plume disperses"

has been changed to:

> "Additionally, there is a known bias in the OMI data due to the limited sensitivity of nadir instruments as the plume disperses" (see lines 401-402 in the revised manuscript).

***l.98: 'high spectral resolution'***

Should this not read 'high spatial resolution'?

 When downloading the data from https://imk-asf-mipas.imk.kit.edu/mipas/, we had to select either "V5 high spectral resolution" or "V5 low spectral resolution". We used the V5 high spectral resolution data.

***l.106: 'are reported on pressure levels with an approximate spacing of...'***

The reported retrieval-grid of remote sensing data is generally not equal to its spatial resolution. Therefore, I would suggest to add the information on resolution from here: https://mls.jpl.nasa.gov/data/v5-0_data_quality_document.pdf, p. 157

Thank you for the suggestion. We updated the resolution numbers and added a citation for the data quality document. See lines 134-136 in the revised manuscript.

***l.146: 'to the pressure coordinate of the MIPAS data'***

Shouldn't this read 'to the altitude coordinate...'?

Yes it should! Thank you for the catch. We changed the text as suggested.

*l.236: 'The reason for this remains unclear, and no explanation or documentation for this difference was found in the literature.'*

You might try to contact the MLS-team for a possible explanation(?)

We have had personal correspondence with Hugh Pumphrey, one of the leading scientists working with MLS. He was also perplexed by the difference, and we did not get a resolution talking with him.

*Chapter 3.2: 'Background seasonal cycle in the MLS data'*

Is there any possibility to infer the disturbing seasonal cycle from comparing different years with less volcanic influence?

Thank you for the question and suggestion. At each day of the year, there is a spread of around 100-200 Gg in the SO2 seasonal cycle during non-volcanic times. Supposing we take the mean of the background years as the reference seasonal cycle, the actual disturbing seasonal cycle could be $\pm$50-100 Gg offset from the reference. This would impart a large bias on the calculated decay rate, thus limiting our ability to constrain the uncertainty.

The interfering background seasonal cycle is interesting in its own right, and to our knowledge under-explored in the literature. Given that it is likely due to $O_3$ and $HNO_3$, perhaps one might be able to calibrate the seasonal cycle based on observed values of these trace gases. However, we feel this would be beyond the scope of the current study. Nonetheless, we have added further discussion on this point (lines 305-312) in Section 4.2 as well as a supplementary figure (S2).

> "The approach outlined above samples possible seasonal cycles using the observed time series in a given year of an eruption. One could also potentially infer the interfering seasonal cycle from years without large volcanic eruptions; however, this presents its own challenges. There is a 100 to 200 Gg spread in $SO_2$ mass on any given day of the year in non-volcanic conditions (Fig. S2). Using the mean across years for the background seasonal cycle could result in an offset of $\pm$50 to 100 Gg from actual disturbing seasonal cycle. This difference is large enough to impart a significant bias in the estimated decay rate, thus limiting our ability to constrain the uncertainty. The seasonal cycle and inter-seasonal variability in MLS $SO_2$ is interesting in its own right: it warrants further investigation and could potentially be calibrated based on the observed amounts of $O_3$ and $HNO_3$ but this is beyond the scope of the current paper."

**l.290, Table 1:**

I strongly recommend consolidating all the information from Table 1 and Table B1 into a single comprehensive table. Additionally, I suggest including the uncertainties reported in Höpfner et al. (2015) and the results from the WACCM simulations, including those for the 18–22 km layer if available. This enhancement would greatly improve the readability of the discussion in Chapter 3.3, allowing readers to follow the analysis more easily without needing to consult multiple tables.

Thank you for the suggestion. We agree this improves readability, and Table 1 and the surrounding discussion in section 4.3 has been updated accordingly

**l.290: 'Their values show a clear increase of e-folding time with height, which is not as apparent in our results.'**

However, the decay timescales provided here also show increasing values (with one exception of MLS in case of Kasatochi). Further, I suggest also to try to provide best-estimates from your analysis of the decay-timescales at 18-22 km for MIPAS in case of Kasatochi and Sarychev – when looking at Fig. 2 and Fig. A1, there seems to be a clear signal.

We adjusted the text following Table 1 to read:

> "Their values show a clear increase of decay rate with height, which is generally also seen in our results (with the exception of some of the values for Kasatochi)" (lines 349 to 351 in the revised manuscript).

We also added MIPAS estimates for Kasatochi and Sarychev in the 18-22 km height bin in Fig. 3, Fig. 5. Fig. A1, and Table 1.

**l.295-314: comparison to WACCM**

The discussion may give the impression that the uncertainties in the decay timescales derived from the measurements are too large to allow for meaningful comparisons with the model. However, upon examining Figures 5, C1, and C2, it appears that for the Sarychev and Nabro eruptions, the model significantly underestimates the decay timescales compared to the limb-sounding datasets, whereas this discrepancy is less pronounced for the Kasatochi eruption. Could you provide possible explanations for this observed difference?

One possible explanation for the observed difference could be the fact that the model puts most of the $SO_2$ from the Kasatochi eruption into the 14-18 km height bin, whereas most of the $SO_2$ for the Sarychev eruption is in the 10-14 km height bin. The model also puts a substantial amount of $SO_2$ in the 10-14 km height bin for the Nabro eruption. As OH decreases with height in the model, $SO_2$ is going to get oxidized and removed more slowly the higher up it is. This could account for the slower decay timescales (and thus closer to MIPAS) reported for Kasatochi (as compared to Sarychev and Nabro) when looking at the larger vertical column. We added a figure (S2) to the supplement showing this and a brief discussion in lines 405-409 in the revised manuscript.

[Figure]

l.355: 'our main focus here is on the decay times of the stratospheric inputs of the indicated eruptions and not the total stratospheric mass.'

The entire chapter 5 of the manuscript (as well as section 2.5 'Calculation of total stratospheric SO2 burden') is dedicated to the 'Estimating the stratospheric SO2 burden'. Therefore, I don't understand this statement. I would suggest to delete this sentence and extend chapter 5 a bit by extending Table 2 to include the estimations of stratospheric SO2 mass by the extrapolation methods used by Pumphrey et al. (2015) and Höpfner et al. (2015). I would also suggest to add the results (M(t0)) from the fits

performed in the present work. It should be made clear that the method described here in section 2.5 is not adequate to calculate the total stratospheric SO2 burden in case of MIPAS and, to a less extend, also for MLS.

We have adjusted the sentence you highlighted to read (lines 465 to 466 in the revised manuscript):

> "The main focus of the paper is on the decay times of the stratospheric inputs from the indicated eruptions, and Table 2 explores the implications of such information for estimating the total stratospheric mass burden."

Your comment about presenting the inadequacy of method presented in Section 2.5 is well taken, and we have adjusted the text in Section 6 to emphasize this more clearly. For example, see lines 428-430 in the revised manuscript. Note we have removed Section 2.5 and included its content in Section 6.

> "As discussed in more detail below, this method proves inadequate for accurate mass burden estimations. This is particularly true for MIPAS, as the spectral bands measured by MIPAS saturate at high $SO_2$ concentrations. The inclusion of it in this paper provides a contrast compared to previously published total burden estimates derived from $SO_2$ decay timescale estimates..."

Finally, we have reworked Section 6 and updated Table 2 to include our own exponential fit-derived estimates of the total $SO_2$ burden. Please see the revised document for the changes.

*l.400: '...and should be considered when analyzing volcanic SO2 with OMI'*

I would suggest to add here: 'and other nadir-sounding instruments'.

Thank you for the suggestion; this text was added.

*l.413-417: eruptions with ash*

The eruption of Puyehue in June 2011 was also rich in ash. Have you tried to inspect that one for any effects on SO2 lifetime? (e.g. Griessbach et al., 2016, doi:10.5194/amt-9-4399-2016)

Thank you for the comment, as it is a good point. We considered other eruptions during the time period of overlap between MLS and MIPAS. However, we focus here on three specific eruptions in this paper because they were large enough to allow for a calculation of the e-folding time of $SO_2$. Puyehue, as well as other notable eruptions during this time period such as Grimsvotn had too weak and noisy of a signal, particularly in the MLS data, to calculate the decay rate. As we are interested in comparing MLS and MIPAS, we leave these smaller eruptions out of the analysis.

*l.424: 'Our work suggests that the current SO2 data reported by available observational products are subject to significant uncertainty when examining the stratospheric lifetime of volcanic SO2 and suggests that more precise data is needed if chemical mechanisms and SO2 mass loading following an eruption are to be elucidated using observed decay times.'*

On one hand, I support this statement, particularly considering the imminent loss of limb-sounding capabilities for stratospheric $SO_2$ observations, which will create a significant gap in our ability to monitor the stratosphere. On the other hand, I find the statement somewhat overly general. As noted in the manuscript, each observational technique has specific advantages and limitations in quantifying stratospheric $SO_2$. Therefore, to effectively evaluate and refine models, it may be more appropriate to tailor comparisons to align with the strengths of each dataset. For example, model results could be compared directly with nadir and MLS data closer to the eruption time, while comparisons with IR limb-sounding datasets might be more suitable for periods several weeks after the eruption.

Thank you for the suggestion. We have modified the last paragraph (lines 523-527 in the revised manuscript) to read as follows:

> "Our work suggests that the current $SO_2$ data reported by available observational products are subject to significant uncertainty when examining the stratospheric decay of volcanic $SO_2$. The varying strengths and shortcomings of the different observational products should be accounted for when using them to determine chemical mechanisms and $SO_2$ mass loading. Furthermore, the forthcoming loss of MLS (the only limb-sounding $SO_2$ instrument in operation) will leave a significant gap in our ability to monitor the stratosphere."

---

## Referee Report (RR1)

Review of "Quantifying the decay rate of volcanic sulfur dioxide in the Stratosphere"

This is my second review of the paper. A number of my recommendations were ignored in this revision.

There are two main issues that need to be addressed.

(1) The authors fail to resolve or even suggest a resolution for the very different SO2 decay rates between total column SO2 (OMI) limb SO2 (MLS/MIPAS) (see Table 2). They need to come up with a plausible explanation for these large differences otherwise this paper is adding noise (not signal) to the issue of SO2 decay.

(2) I also have a serious comment on the interpretation of the exponential decay of SO2 based on the recent paper by Toohey et al. (2025; https://doi.org/10.5194/acp-25-3821-2025). SO2 can decrease due to two processes in the 10-22 km region: (1) conversion to sulfate aerosol and (2) transport out of the stratospheric domain. These processes have different time scales. The closer the eruption is to the troposphere boundary, the dynamical transport will accelerate the loss - assuming that SO2 lifetime in the troposphere is much shorter than in the stratosphere (about 2 weeks, Beirle et al., 2014). This dynamical effect will appear as a faster exponential decay which the authors attribute to stratospheric chemistry. To give an example, let us say that an eruption takes place close to the tropopause – if the gas were neutral in the stratosphere and the loss was only in the troposphere, the stratospheric lifetime might be ~2 months. If we fold in the SO2 actual chemical decay time in the stratosphere (say 1 month McKeen et al., 1984) then the observed lifetime would be ~20 days. In other words, the observed lifetime of SO2 is a mix of chemical and dynamical lifetimes whereas here it is interpreted as purely chemical.

Specific comments are below.

Ln 12 It would be useful to the reader to indicate the range of lifetimes rather than leave it as "difficult to attribute"

Ln 30 'once the plume reaches the stratosphere'

Ln 41 There a lot more volcanic aerosols impacting climate references (see Robock, Stenchikov, etc.)

In the literature review, you should cite McKeen et al. (1984) here as well as in line 173

Ln 90 Is there a reason TROPOMI wasn't used? Its spatial resolution is higher than OMI?

Ln 107 Livesey is spelled wrong.

Ln 112 Actually, OMI does provide ozone profiles using the SBUV profiler methodology so what you say here is not quite true.

Ln 115 The Dobson unit definition is wrong (cm$^{-2}$)

Ln 151 I asked this question in a previous review, why not divide that atmosphere from 10-14 km, 14-18 km and 18-22 km. The lower bins include the troposphere in the tropics which is a quite different region than the stratosphere.

Ln 198 Aura is not in a decaying orbit.  To preserve fuel, Aura is not following Aqua and so the crossing time has drifted. With current fuel reserves, Aura (and MLS) will last until 2028 after which power levels are too low to run the instrument.

Ln 218 Do the V2 MLS SO2 retrieval problems persist in V5?  See more about this below.

Ln 236, 243 The MLS data quality document clearly discusses the SO2 changes from early versions. There is no mystery here. The MLS user guide can be downloaded from the JPL website.  Negative mixing ratios are an artifact of the retrieval. They are not bad data, and indicate that averaging over a larger volume is required.

Ln 250  I don't understand why you think that the seasonal cycle is unrealistic. Transport from the troposphere to the stratosphere has a seasonal cycle so I expect that upward transport of OCS and total SO2 would have a seasonal cycle.

Table 1. It would be helpful if you added latitude, longitude and exact eruption date next to the volcano name. If you ignore the 10-14 km range, where the decay rate may be accelerated due to dynamics, the numbers are in reasonable agreement.  Also, the uncertainty of the MLS measurements is much higher between 8 and 12 km.  The high uncertainty value for NABRO MLS SO2 seems a little weird to me.  The 14-18 range is in the upper troposphere so I am not sure you aren't getting good data.

Fig. A2  What is producing the spikes about day 50 and 60? Those spikes are absent from MIPAS data.

The increase in e-folding times with height found you found and also in Höpfner et al. is, I believe due to dynamics – transport across the tropopause is weaker at higher altitudes. This issued is mentioned in the beginning of the review.

Ln 320 You should fix line 112 to be consistent with this statement.

Ln 330 Another explanation for the differences between OMI and limb sounders is that

OMI is losing mass due plume dispersal. As the plume spreads out, the pixels with smaller amounts of SO2 will no longer register and thus the plume would "appear" lose SO2 when it is (in fact) not – the plume edges have fallen below the detection limit. The limb sounders also face this problem if they fail to acquire a plume on successive orbits, but since a significant amount of the OMI plume is in the upper troposphere it is likely worse.

Table 2 The large SO2 differences between OMI and the other instruments for Kasatochi is not explained and needs to be.

Ln 384 The version of MLS you are using, I believe, corrects for HNO3 and O3 interference. You need to reference and discuss the MLS V5 documentation of the SO2 retrieval.

Ln 398 We are left hanging on the OMI vs MLS differences in decay rate.

Ln 418 Please re-label HTHH as Hunga consistent with the community recommendation.

---

## Author Response (AR3)

**Quantifying the decay rate of volcanic sulfur dioxide in the stratosphere**

Response to reviewers and editors in red. All line numbers refer to the revised manuscript.
* * *
**Editor**

Furthermore, please revise based on the following two comments:
line 41: "Once formed, stratospheric sulfate aerosols have a residence time of 1 to 2 years (Kremser et al., 2016)." First, this estimate of residence time is valid only for large tropical eruptions--many references will quote a different residence time for high latitude eruptions, which is relevant to your study since two of the eruptions you focus on are high latitude. Secondly, I cannot find this precise statement in the paper by Kremser et al. (2016), although it is common in other references.

Thank you for the comment. We have updated the language to be more precise and also added more appropriate references. Please see lines 43-46 in the revised manuscript:

> "Once formed, the residence time of stratospheric sulfate aerosols ranges from a few months to a couple years and depends on the latitude, injection height, and time of year of the eruption. High latitude eruptions with relatively low injection heights are associated with shorter residence times, whereas the aerosol cloud from tropical eruptions with high injection heights can persist for 1 to 2 years (Toohey et al., 2025; Myhre et al., 2013)."

line 224: Here you claim similarity between the decay timescale of SO2 and the timescale of the increase in sulfate aerosol mass, but the latter seems that it would be very sensitive to the period chosen to perform the fit over. How did you choose the particular period for the fit, and how sensitive is your conclusion to the choice of period?

Thank you for the comment. We have elaborated on this comment in Section 3. Please see the revised manuscript (lines 230-255).
* * *
**Reviewer #1**

l. 227 (of the file egusphere-2024-3525-manuscript-version3.pdf), 'high spectral resolution':

I'm convinced that this should read 'reduced spectral resolution' since, as it is correctly stated in line 125: 'the spectral resolution was 0.0625 cm−1' and this is the reduced spectral resolution of MIPAS which is valid for the years 2005-2012.

Thank you for bringing this up again, and you are most definitely correct. We apologize for the mistake! The wording has been fixed in the newest version of the manuscript.
* * *
**Reviewer #2**

Review of "Quantifying the decay rate of volcanic sulfur dioxide in the Stratosphere"

This is my second review of the paper. A number of my recommendations were ignored in this revision.

Thank you for taking the time to review the paper again. The authors would like to clarify that we addressed all of your initial comments during the first round of revisions. These are contained in egusphere-2024-3525-author_response-version2.pdf (found in the MS records), as well as in the interactive discussion section. Furthermore, based on the line numbers referenced in the reviewer's comments, it appears the reviewer read the initial submission of the paper as opposed to the revised one.

There are two main issues that need to be addressed.

(1)     The authors fail to resolve or even suggest a resolution for the very different SO2 decay rates between total column SO2 (OMI) limb SO2 (MLS/MIPAS) (see Table 2). They need to come up with a plausible explanation for these large differences otherwise this paper is adding noise (not signal) to the issue of SO2 decay.

The authors believe this text in Section 4.3 addresses this concern (lines 435-440):

> "There are a couple of possible explanations for this. While the OMI data used here is designed to give an estimate of $SO_2$ mass in the stratosphere (Sect. 2.1.3) there is potential for tropospheric $SO_2$ to influence this measurement. Tropospheric $SO_2$ will generally get removed much quicker than that in the stratosphere, and could be

skewing the decay rates reported here. Additionally, there is a known bias in the OMI data due to the limited sensitivity of nadir instruments as the plume disperses (see Sect. 2.1). Both of these should be considered more carefully when analyzing OMI $SO_2$ following an eruption."

Additionally, we have added the following in the conclusion (lines 534-536):

"This is a bias, perhaps due to interference from tropospheric $SO_2$ and nadir instruments' limited sensitivity to dispersed plumes. It should be considered when analyzing volcanic $SO_2$ with OMI and other nadir-sounding instruments."

(2)      I also have a serious comment on the interpretation of the exponential decay of SO2 based on the recent paper by Toohey et al. (2025; https://doi.org/10.5194/acp-25- 3821-2025). SO2 can decrease due to two processes in the 10-22 km region: (1) conversion to sulfate aerosol and (2) transport out of the stratospheric domain. These processes have different time scales. The closer the eruption is to the troposphere boundary, the dynamical transport will accelerate the loss - assuming that SO2 lifetime in the troposphere is much shorter than in the stratosphere (about 2 weeks, Beirle et al., 2014). This dynamical effect will appear as a faster exponential decay which the authors attribute to stratospheric chemistry. To give an example, let us say that an eruption takes place close to the tropopause – if the gas were neutral in the stratosphere and the loss was only in the troposphere, the stratospheric lifetime might be ~2 months. If we fold in the SO2 actual chemical decay time in the stratosphere (say 1 month McKeen et al., 1984) then the observed lifetime would be ~20 days. In other words, the observed lifetime of SO2 is a mix of chemical and dynamical lifetimes whereas here it is interpreted as purely chemical.

Thank you for this comment. However, the authors argue that vertical transport will be a higher order effect on the decay rates calculated here and thus only have a minor impact. For instance, the lifetime of sulfate aerosols formed after volcanic eruptions is on the order of several months to a couple years (Toohey et al., 2025, Myhre et al., 2013), which is appreciably longer than the timescales for $SO_2$ oxidation by OH; these aerosols are removed not only by vertical advection, but also sedimentation (which results in a faster aerosol removal than could be done by advection alone). Thus, we believe (in agreement with previous literature) that the variation of decay timescales with height are primarily due to chemical processes.

To hopefully clarify some of the confusion around the role of vertical advection, we have added the following discussion to the paper (lines 371-376)

"Vertical transport by the background circulation of the stratosphere is unlikely to have a significant impact on our results as it is quite slow---on the order of tenths of mm/s or hundreths of km/day---compared to the timescale of $SO_2$ decay (Butchart, 2014). Khaykin et al., (2022) did report an unusual radiative self-lofting of the Raikoke volcanic plume in 2019; the observed vertical ascent for this eruption was upwards of 2 mm/s

(0.17 km/day) and would be fast enough to impact our results. This phenomenon has not been noted for any of the volcanoes examined here, though it is a potential source of uncertainty and worth examining in future work."

Specific comments are below.

Ln 12 It would be useful to the reader to indicate the range of lifetimes rather than leave it as "difficult to attribute"

Thank you for the suggestion. We have changed the text to read:

"While the typical decay timescale for SO2 is on the order of a few weeks to a month, we find that uncertainties across different altitudes and eruptions results in lifetimes that can vary by more than a factor of 2. This makes it difficult to attribute variations in decay timescale to specific SO2 removal processes for the events examined."

Ln 30 'once the plume reaches the stratosphere'

We have changed the wording as suggested.

Ln 41 There a lot more volcanic aerosols impacting climate references (see Robock, Stenchikov, etc.)
In the literature review, you should cite McKeen et al. (1984) here as well as in line 173

We added references to Robock (2000), Stenchikov et al., (2009) to the statement about aerosols impacting climate. Additionally, we added the McKeen et al., (1984) citation to the recommended location.

Ln 90 Is there a reason TROPOMI wasn't used? Its spatial resolution is higher than OMI?

TROPOMI data begins in 2018 and thus doesn't cover the eruptions that we focus on in this paper, whereas OMI does. Furthermore, while TROPOMI does have higher resolution, we choose to focus on OMI as there were several papers that use OMI to derive $SO_2$ decay rates that motivated this paper initially (see, for example, Zhu et al., (2020), Zhu et al., (2022), and Krotkov et al., (2008)).

Ln 107 Livesey is spelled wrong.

Thank you for the catch! The spelling has been corrected.

Ln 112 Actually, OMI does provide ozone profiles using the SBUV profiler methodology so what you say here is not quite true.

We have updated the text to specify that OMI does not provide high-resolution vertical resolution for $SO_2$.

Ln 115 The Dobson unit definition is wrong (cm $^{-2}$)

Thank you for pointing this out. The unit has been corrected.

Ln 151 I asked this question in a previous review, why not divide that atmosphere from 10- 14 km, 14-18 km and 18-22 km. The lower bins include the troposphere in the tropics which is a quite different region than the stratosphere.

This is an important point, and one that we addressed in our response to your initial review of the manuscript. The line number referenced does not align with the location of this discussion in the revised manuscript, which perhaps suggest that our response was missed. Since we have already addressed this point, I have copied the previous response below:

Thank you for the comment. The main rational behind using the three layers is that it allows for comparison to previous work by Höpfner et al., (2015). We agree, however, that whether or not these layers fall in the stratosphere is going to be highly latitude dependent. As such, we have included a more detailed discussion of where these layers fall as a function of time-of-year and latitude in the beginning of Section 4.3 (lines 354 to 359 in the revised manuscript):

> "In particular, the tropopause in the tropics during the local summer is around 16 km, whereas that for the high northern hemisphere latitudes is closer to 11km (Hoffmann and Spang, 2022). As such, the majority of the three layers considered in this analysis are likely to be in the stratosphere for the Kasatochi and Sarychev eruptions. After the Nabro eruption, likely only the 18 to 22 km layer was initially fully in the stratosphere; however, the plume was quickly advected to higher latitudes—where the tropopause is lower—by the Asian Monsoon anticyclone in just a few days (Clarisse et al., 2014)."

We think that in order to facilitate comparison with past work, keeping the analysis focused on these three layers is the best choice. However, we have added a comment on the validity of this choice in light of the concerns you raise. See the updated text in Section 2.3 (lines 193 to 196) in the revised manuscript):

> "Given the variation in tropopause height with latitude, the 10 to 14 km and 14 to 18 km layers won't necessarily be entirely in the stratosphere in low latitudes (Hoffmann and

Spang, 2022). However, we use the vertical divisions here for consistency with past work, and future work could consider a division based on tropopause height."

Ln 198 Aura is not in a decaying orbit. To preserve fuel, Aura is not following Aqua and so the crossing time has drifted. With current fuel reserves, Aura (and MLS) will last until 2028 after which power levels are too low to run the instrument.

We have refined the language to read as:

"The Aura satellite, which carries the MLS instrument, is expected to last until 2028, while MIPAS operated from 2002 through 2012."

Ln 218 Do the V2 MLS SO2 retrieval problems persist in V5? See more about this below.

Yes, based on our Figure 3, for example, it appears similar problems with regards to the seasonal cycle persist in V5. See the following comment for more details and modifications that we have made to the text.

Ln 236, 243 The MLS data quality document clearly discusses the SO2 changes from early versions. There is no mystery here. The MLS user guide can be downloaded from the JPL website. Negative mixing ratios are an artifact of the retrieval. They are not bad data, and indicate that averaging over a larger volume is required.

We have downloaded and read through the MLS data quality document thoroughly. The changes stated in the data quality document for V5 mention changes in the channels used for $O_3$ and CO lines, which "will have secondary impacts on $SO_2$." Additionally, the documentation highlights that all versions of the MLS dataset are biased high "due to systematic errors in the MLS measurement system." There is no mention of the seasonal cycle or the change in magnitude between V2 and V5 that we report here in lines 308-310.

Furthermore, while the documentation does indeed mention that negative mixing ratios are a by-product of the retrieval, our results here show that these negative biases persist even when the data is aggregated over a large area (40°N-90°N, for example).

We have modified the text in lines 277 to 280 to make the reference to the documentation more explicit:

"Negative mixing ratios are unphysical and an artifact of the MLS retrieval algorithm; nonetheless, the MLS documentation recommends including them in subsequent calculations (Livesey et al., 2022). However, aggregating these negative mixing ratios over a large geographic area (e.g., 40N to 90N) and converting to mass results in large negative values that complicate the interpretation of the data, but are nonetheless included here."

We have also added some additional context to the MLS dataset in section 2.1.2 (lines 143 to 153):

> "We use Level 2 Version 5 (V5) daily swath $SO_2$ mixing ratio data, accessed at https://disc.gsfc.nasa.gov/datasets/ML2SO2005 (Read and Livesey, 2021). The V5 data features minor changes from previous versions, including improved cloud detection, changes in the calculation of $O_3$ and carbon monoxide (stated to have secondary impacts on $SO_2$), and updates to the handling of background radiance signals (Livesey et al., 2022). This data is obtained via the 240 GHz radiometer on the MLS instrument (Pumphrey et al., 2015). In addition to the $SO_2$ mixing ratio, this dataset reports the temperature at each pressure level, and we use this in our calculation of $SO_2$ mass and altitude above sea level.
>
> The MLS documentation highlights that the retrieval algorithm can generate negative mixing ratios, and the correct way to deal with these is to average over a sufficiently large horizontal area (Livesey et al., 2022). We apply all of the suggested masking for the data given in Livesey et al., (2022), and we average our data over 10° latitude bands. Even after masking and averaging, negative mixing ratios are prominent in the MLS data, particularly lower in the atmosphere."

Ln 250  I don't understand why you think that the seasonal cycle is unrealistic. Transport from the troposphere to the stratosphere has a seasonal cycle so I expect that upward transport of OCS and total SO2 would have a seasonal cycle.

It is not so much that the seasonal cycle is unrealistic, but the amplitude of the of the seasonal cycle seen here (~100 $GgSO_2$), particularly in the lower height bins, is too large to be explained by OCS transport or any other reasonable source of $SO_2$ in the stratosphere. Data from Höpfner et al., (2013) shows background $SO_2$ in the lower stratosphere on the order of only a few ppt (their figure 7), whereas the seasonal cycle of $SO_2$ from MLS has an amplitude on the order of ppb (see also Pumphrey et al., (2015)), which further suggests the seasonal cycle is unrealistic.

We have added some clarifying language about the MLS seasonal cycle in lines 284 to 289:

> "Additionally, the MLS mass in the 10 to 14 km and 14 to 18 km bins feature a seasonal cycle with an amplitude much larger than what is expected for background stratospheric $SO_2$ (Pumphrey et al., 2015; Höpfner et al., 2013). The background $SO_2$ values in this region of the stratosphere are on the order of a few tens to a hundred ppt (see Fig. 7 in Höpfner et al., 2013), whereas the seasonal cycle shown here and in Pumphrey et al. (2015) have an amplitude on the order of ppb. Furthermore, the amplitude of the MLS seasonal cycle is too large to be explained by other potential sources of stratospheric sulfur such as the annual flux of OCS into the stratosphere (Karu et al., 2023)."

Table 1. It would be helpful if you added latitude, longitude and exact eruption date next to the volcano name. If you ignore the 10-14 km range, where the decay rate may be accelerated due to dynamics, the numbers are in reasonable agreement.  Also, the uncertainty of the MLS measurements is much higher between 8 and 12 km. The high uncertainty value for NABRO MLS SO2 seems a little weird to me. The 14-18 range is in the upper troposphere so I am not sure you aren't getting good data.

Thank you for the suggestion. We have added dates and geographic coordinates to both Tables 1 and 2.

As you highlight below, the MLS Nabro data is quite noisy, which leads to the large range for the 14-18 km decay rate reported in the table.

Fig. A2  What is producing the spikes about day 50 and 60? Those spikes are absent from MIPAS data.

It is not clear to the authors what these spikes are. Given that the explosive activity from the 2011 Nabro eruption had ceased by this time (see the eruption timeline from the Smithsonian Global Volcanism Program), we simply treat these spikes as noise in the signal.

There were a few other notable eruptions in 2011 after Nabro. Soputan erupted in Indonesia in early July of that year. Etna (Italy) and Hudson (Chile) erupted in October 2011. However, the timing of these eruptions is such that they cannot explain the spikes seen in the figure.

The increase in e-folding times with height found you found and also in Höpfner et al. is, I believe due to dynamics – transport across the tropopause is weaker at higher altitudes. This issued is mentioned in the beginning of the review.

Please see our response to comment (2) above.

Ln 320 You should fix line 112 to be consistent with this statement.

We have changed the wording to read (lines 422-423):

> "OMI reports the vertical column density of $SO_2$ and lacks the explicit vertical resolution for $SO_2$ provided by MLS and MIPAS."

Ln 330 Another explanation for the differences between OMI and limb sounders is  that OMI is losing mass due plume dispersal. As the plume spreads out, the pixels with smaller amounts of SO2 will no longer register and thus the plume would "appear" lose SO2 when it is (in fact) not – the plume edges have fallen below the detection limit.  The limb sounders also face this problem if they fail to acquire a plume on successive orbits, but since a significant amount of the OMI plume is in the

upper troposphere it is likely worse.

We have added the following text in lines 435 to 440 in the revised manuscript:

> "There are a couple of possible explanations for this. While the OMI data used here is designed to give an estimate of $SO_2$ mass in the stratosphere (Sect. 2.1.3) there is potential for tropospheric $SO_2$ to influence this measurement. Tropospheric $SO_2$ will generally get removed much quicker than that in the stratosphere, and could be skewing the decay rates reported here. Additionally, there is a known bias in the OMI data due to the limited sensitivity of nadir instruments as the plume disperses (see Sect. 2.1). Both of these should be considered more carefully when analyzing OMI $SO_2$ following an eruption."

Table 2 The large SO2 differences between OMI and the other instruments for Kasatochi is not explained and needs to be.

The authors also find this odd, and we are not sure why there is such a large difference for Kasatochi when compared to the other eruptions analyzed. We did modify the language in lines 490 to 492 to address this:

> "We also note that the MIPAS and MLS $SO_2$ masses in some eruptions are significantly lower compared to the Carn (2024) values. Whether this reflects fractional stratospheric inputs or biases due to limitations of sampling by limb-sounding instruments would be a subject for future research."

Ln 384 The version of MLS you are using, I believe, corrects for HNO3 and O3 interference. You need to reference and discuss the MLS V5 documentation of the SO2 retrieval.

We added some relevant comments and references to the MLS documentation to Section 2.1.2 (see lines 143 to 153 in the revised manuscript):

Ln 398 We are left hanging on the OMI vs MLS differences in decay rate.

Please see the reply to comment (1) above.

Ln 418 Please re-label HTHH as Hunga consistent with the community recommendation.

Based on the previous review, we have already changed this wording as suggested.

---

## Author Response (AR4)

**Quantifying the decay rate of volcanic sulfur dioxide in the stratosphere**

Thank you for the detailed comments on the paper. Our response to these comments in below in red. All line numbers refer to the revised manuscript.
* * *
Minor comments:

Fig 1 caption: please modify the figure legend text to reflect the manuscript's use of the term "decay timescale" rather than "e-folding time".

The figure has been updated.

Line 240: If M1 is not equal to M0, then this implies that at t=0, there is a non-zero amount of sulfate aerosol. Is this an intentional choice, to represent some directly injected aerosol? Otherwise, a normal procedure would assume that the mass of aerosol is zero at t=0, which provides a boundary condition that requires M1=M0. This provides a cleaner expression for sulfate aerosol mass, and perhaps the results of the analysis would be identical as presented here. Please consider, but if you keep the present form, a few more words on the physical meaning of M1 and M0 would be useful to include.

Thank you for raising this point. We were not trying to represent any directly injected aerosol, and we agree that having $M_1 = M_0$ allows for a cleaner expression for the sulfate aerosol. We have updated the equations in the manuscript to reflect this (Line 240). This update in notation doesn't have any impact on the results of the analysis.

Line 243: Don't you use a log fit here (or a linear fit to the log(Mso4) time series)?!

This has been clarified in line 243-244. The text now reads:
> "We then take the natural log of the resulting curve and use a linear fit to estimate $\tau$."

Line 563: This is not actually accurate, since the ACE-FTS instrument measures SO2 and is currently operating. See e.g., Cameron et al., 2021). Please modify this statement.

Cameron, W. D., Bernath, P., and Boone, C.: Sulfur dioxide from the atmospheric chemistry experiment (ACE) satellite, J. Quant. Spectrosc. Radiat. Transf., 258, 107341, https://doi.org/10.1016/j.jqsrt.2020.107341, 2021.

This text has been updated as follows (Line 563):
> "Furthermore, the forthcoming loss of MLS (the only limb-sounding $SO_2$ instrument in operation with continuous coverage over global latitudes and longitudes) will leave a significant gap in our ability to monitor the stratosphere."